# Graph Neural Networks for Road Safety Modeling: Datasets and Evaluations for Accident Analysis

**Abhinav Nippani[‡] , Dongyue Li[‡], Haotian Ju, Haris N. Koutsopoulos, Hongyang R. Zhang**
Northeastern University, Boston
{nippani.a, li.dongyu, ju.h, h.koutsopoulos, ho.zhang}@northeastern.edu

## Abstract

We consider the problem of traffic accident analysis on a road network based on road network connections and traffic volume. Previous works have designed various deep-learning methods using historical records to predict traffic accident occurrences. However, there is a lack of consensus on how accurate existing methods are, and a fundamental issue is the lack of public accident datasets for comprehensive evaluations. This paper constructs a large-scale, unified dataset of traffic accident records from official reports of various states in the US, totaling 9 million records, accompanied by road networks and traffic volume reports. Using this new dataset, we evaluate existing deep-learning methods for predicting the occurrence of accidents on road networks. Our main finding is that graph neural networks such as GraphSAGE can accurately predict the number of accidents on roads with less than 22% mean absolute error (relative to the actual count) and whether an accident will occur or not with over 87% AUROC, averaged over states. We achieve these results by using multitask learning to account for cross-state variabilities (e.g., availability of accident labels) and transfer learning to combine traffic volume with accident prediction. Ablation studies highlight the importance of road graph-structural features, amongst other features. Lastly, we discuss the implications of the analysis and develop a package for easily using our new dataset.

## 1 Introduction

Graph neural networks are widely used tools for extracting structural relationships from data. Examples include friendships and interactions on social networks [34, 10], 3D protein-protein interactions [61, 45], and road connections on traffic networks [39]. Motivated by the widespread use of graph neural networks, large-scale graph databases and benchmarks have received significant interest in recent studies [26, 14, 25]. Existing architecture designs can be abstracted in the mathematical framework of message-passing neural networks [63]. In practice, the empirical performance of different network designs varies across domains [12]. To facilitate the discussion, this paper examines graph neural networks for the important problem of traffic accident risk modeling: Given historical accident accords as edge labels on a road network, how well can we learn to predict the number of accident occurrences on roads (e.g., over the next month) using graph-structural and related features?

The importance of modeling traffic accident risks is well-recognized as many counties [46, 3, 5, 43] propose vision zero plans to eliminate motor vehicle crashes. According to CDC [56], the economic cost of crashes amounts to $430 Billion in 2020 [9]. Developing better modeling tools can identify the underlying risk of accidents at a certain location [23], thus informing policy intervention [2].

Many studies have analyzed the effect of road features for predicting accident occurrences, such as traffic flow, road network geometry, and rain [49, 15, 18]. Caliendo et al. [6] conduct a regression

---

[‡]First two authors contributed equally.

37th Conference on Neural Information Processing Systems (NeurIPS 2023) Track on Datasets and Benchmarks.

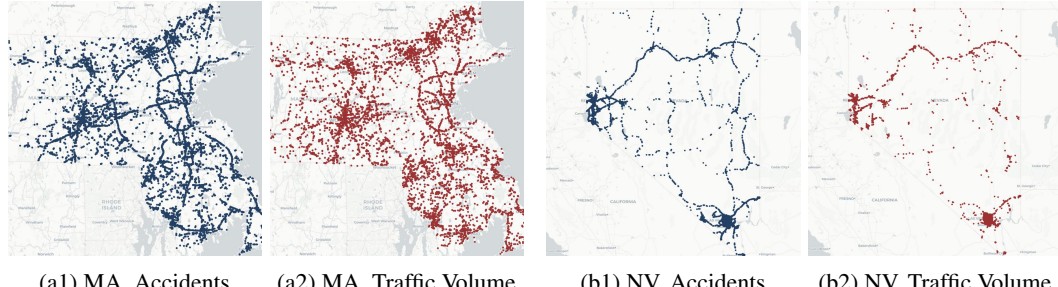

| (a1) MA, Accidents | (a2) MA, Traffic Volume | (b1) NV, Accidents | (b2) NV, Traffic Volume |

Figure 1: We note there is a clear association between accident occurrences and traffic volume. Combining accident and annual average daily traffic reports using transfer learning techniques can improve accident prediction by 4.6%. Further, road network structural features across states are most predictive of accident occurrences. We capture cross-state variability using multitask learning by combining the labels from all states. This allows information transfer from states with rich data to states with fewer labels. We find that this outperforms learning from individual state data by 4.7%.

analysis to quantify the safety effects of the annual average daily traffic (AADT) and rain (among others) using observed crash data in Italy. Ihueze and Onwurah [29] incorporate human mobility factors into road accident modeling with autoregressive models. These studies focus on simple regression models. The use of deep learning for traffic accident prediction has recently been examined [41, 52, 73]. Yuan et al. [70] develop a heterogeneous convolutional LSTM framework to predict traffic accidents based on data collected from Iowa state. Moosavi et al. [44] collect over 2 million accident records and develop feedforward networks to predict accident labels on road networks. Despite significant recent interests and strong societal importance, it remains unclear how accurately existing deep learning methods can be used to predict accident occurrences on road networks.

A critical challenge in addressing this question is a lack of (large-scale) traffic accident datasets. There is a large body of work focusing on traffic forecasting (see, e.g., Li et al. [39], Wang et al. [58], and Jiang and Luo [31]), and the datasets therein are usually not annotated with accident records. Moosavi et al. [44] construct a repository including over 2 million accident records collected from map APIs across US states. By contrast, we construct a new dataset with over 9 million traffic accident records spanning eight states, the longest spanning twenty years. We extract these records from official reports of the Department of Transportation from each state, each comes with the latitude, longitude, and time of day of occurrence. This is a nontrivial task as different states publish their data under different formats and APIs, and our dataset unifies them all into the same format. We then collect road networks, road-level AADT, and weather reports aligned with the accident labels in our dataset.

Based on this new dataset, we evaluate existing neural network models in terms of their performance in predicting accident occurrences. Our major finding is that using road structural features and traffic volume reports, existing graph neural networks such as GraphSAGE [20] can predict the accident counts with **22%** mean absolute error (relative to actual counts) and whether an accident occurs on the road with over **87%** AUROC, averaged over eight states. We achieve this result by developing multitask and transfer learning techniques on top of the graph neural network, inspired by recent developments in this space [32, 35, 36]. Interestingly, we notice strong cross-sectional trends regarding graph structural patterns across states, as illustrated in Figure 1. As a remark, we clarify that the results should be interpreted as showing that simple graph neural networks achieve comparable performance to (if not outperforming) their more sophisticated counterparts. Part of the reason is that our dataset is sparsely labeled, with a labeling rate of less than 0.3% for all states (cf. Table 2), whereas complex models employ a few times more parameters, making them more challenging to fit. One limitation of our analysis is that the road networks, which we extract from OpenStreetMap [19, 4] make simplifying assumptions without considering traffic flows in sequences of connected road segments with typical routes and multi-lanes. We believe our methodology would extend to this case, e.g., by combining our accident labels with satellite image-based construction of road networks [23].

To summarize, this paper makes three contributions to traffic accident analysis by learning road network structures. First, we construct a unified traffic accident dataset extracted from official reports of eight states, totaling 9 million records, the largest dataset of this kind to our knowledge. Second, we

find that by using road network structures and traffic flow reports, graph neural networks can accurately predict accident labels, suggesting the validity of our approach. Third, we discuss the implications of our analysis and develop a package for easily reusing our new datasets and codes. Our datasets and experiment codes can be found at `https://github.com/VirtuosoResearch/ML4RoadSafety`.

## 2    Methodology

This section presents our approach to collecting and modeling accident data. First, we introduce the problem setup. Second, we describe the collection of a new dataset. Lastly, we present our modeling approach based on graph neural networks, multitask, and transfer learning.

### 2.1    Problem setup

We study the problem of predicting accidents on a road network, viewed as a directed graph $G = (V, E)$, where $V$ denotes a set of road intersections and $E$ denotes a set of roads that connect one intersection to another intersection. Each node $v \in V$ has a list of node features, including the number of incoming and outgoing edges, its betweenness centrality, weather information of the corresponding district, etc. An edge $e \in E$ has features such as the road length, residential or highway category, etc.

Each accident is associated with an edge of the road network where the accident happened, as well as the timestamp during which the accident happened. Given accident records up to a certain time period (e.g., month), we consider the prediction of accident records for the remaining time periods (e.g., months). We measure the result as a regression task by predicting the number of accidents per edge and as a classification task by predicting if one (or more) accidents will occur.

### 2.2    A unified dataset of crashes and road features

**Accident data.** Carrying out machine learning for accident analysis requires collecting historical accident information and predictive features. There are several widely used traffic network datasets, such as METR-LA and PEMS-Bay [30, 39]. Note that these curated datasets do not contain vehicle crash information. There are also online data sources that provide available records for the US [11]. We note that the dataset is collected from streaming APIs that only provide accident information for certain times of the day (e.g., during rush hours) [44]. Further, there is some discussion that the data involves reporting errors in the start and end time [11]. Thus, to ensure the validity of our analysis, we start by collecting data from the official reports of the Department of Transportation and note that several states publish detailed information online, including the latitude and longitude of an occurrence. However, extracting this information is nontrivial: Different states provide the data under different formats and interfaces (some in PDF files). It is not obvious how to combine all these records in a unified format. Thus, our first task is constructing a unified dataset with these records.

To this end, we collected over 9 million records spanning eight states in the US. We provide the basic statistics of this dataset in Table 1. We report our statistics at the state level, including the reported number of accidents per million vehicle millage traveled per year, the number of monthly accidents per state, etc. However, it is possible to carry out this analysis at the county level: Some counties, such as New York City and Los Angeles, report traffic accident information within the county (see links in Table 6, Appendix A). For other details of the collection process, see Appendix A.

**Annual average daily traffic (AADT) reports.** Besides, we explore traffic flow, which is related to crash frequency. The AADT measures the number of vehicles traveling on a particular road. This feature has been known to affect prediction in classical works that apply regression analysis on crash data [49, 15, 18, 6]. However, collecting this data is nontrivial and has not been done in prior works. We collect official reports of *road-level* AADT from the Department of Transportation, which publishes this information under the name of each street; We map the street names to the edges by extracting a coordinate (using Google Map API) and then aligning it to our graph.

**Road features.** For each state, we generate its road network based on OpenStreetMap [19, 4]. Note that this protocol has been widely used in prior traffic forecasting studies (e.g., Li et al. [38], Geng et al. [16], and He et al. [22]). In Table 1, we can see that these road networks are very sparse. Besides the graph, we have also collected many other features to help with prediction. For each road, we have its road category and length information. There are 24 categories in total, such as one-way, highway,

Table 1: Below are the statistics of collected traffic accidents and features in eight states. We calculate crash rate as the number of accidents per vehicle millage traveled (VMT) per year using reported millage values from the corresponding state. We use $d_{avg}$ and $d_{max}$ to denote the average and maximum degree of a graph, respectively. The volume percentage refers to the fraction of roads for which traffic volume reports are available.

| | Start | End | Crash Rate | # Nodes | # Roads | # Accidents/Month |
|---|---|---|---|---|---|---|
| Delaware | 2009 | 2022 | 3.27 | 49,023 | 116,196 | 26,725 |
| Iowa | 2013 | 2022 | 4.92 | 253,623 | 707,072 | 49,495 |
| Illinois | 2012 | 2021 | 36.7 | 627,661 | 1,647,614 | 230,666 |
| Massachusetts | 2002 | 2022 | 24.48 | 285,942 | 706,402 | 70,640 |
| Maryland | 2015 | 2022 | 11.44 | 250,565 | 580,526 | 87,079 |
| Minnesota | 2015 | 2023 | 5.39 | 383,086 | 979,259 | 48,963 |
| Montana | 2016 | 2020 | 1.69 | 145,525 | 351,516 | 17,576 |
| Nevada | 2016 | 2020 | 5.42 | 121,392 | 292,674 | 35,121 |

| | $d_{avg}$ | $d_{max}$ | Centrality ($10^{-3}$) | Avg Length (m) | Volume (%) |
|---|---|---|---|---|---|
| Delaware | 2.4 | 6 | 5.7 | 213 | 3.14 |
| Iowa | 2.8 | 7 | 1.4 | 532 | - |
| Illinois | 2.6 | 8 | 0.8 | 307 | - |
| Massachusetts | 2.5 | 8 | 0.9 | 188 | 1.34 |
| Maryland | 2.3 | 8 | 1.0 | 211 | 1.76 |
| Minnesota | 2.5 | 8 | 0.6 | 474 | - |
| Montana | 2.4 | 7 | 0.02 | 859 | - |
| Nevada | 2.4 | 6 | 0.4 | 280 | 1.38 |

residential, and so on (cf. Appendix A). We encode the physical length in meters as a real-valued feature. For each intersection, we compute its in-degrees, out-degree, and betweenness centrality (which measures the ratio of shortest paths between all node pairs that contain a node), apart from its latitude and longitude. These are the static features. Next, we collect historical (daily) weather information for each node, which is temporal in nature. These include the maximum, minimum, and average temperature, the total precipitation of rainfall and snowfall, the average wind speed, and the sea level air pressure. We align each node to the nearest meteorological station to the node.

### 2.3 Graph neural network modeling for accident prediction

**Graph neural networks (GNN).** The basic unit for our predictive analysis is graph neural networks: Given a graph $G = (V, E)$, along with node-level and edge-level features, a graph neural network applies a neighborhood aggregation mechanism through the edges $E$. Let $l$ be the number of layers. A GNN recursively computes the representations of a node for $l$ layers by aggregating the representations from its neighbors. Let $x_i^{(k)}$ denote the node feature at node $i$ in layer $k$ and $v_{i,j}$ denote the features of edge $(i, j)$. The $k$-th layer of a GNN aggregates note $i$'s neighborhood embeddings to output:

$$x_i^{(k+1)} = \phi\left(x_i^{(k)}, h\left(\left\{\psi\left(x_i^{(k)}, x_j^{(k)}, v_{i,j}\right) : j \in N(i)\right\}\right)\right), \text{ for any } i \in V, \tag{1}$$

where $h$ denotes an aggregation function (e.g., element-wise sum, mean, and max) and $N(i)$ is a set of nodes adjacent to $i$, $\phi$ and $\psi$ denote neural networks with a nonlinear map such as ReLU.

**Cross-state analysis.** Next, we develop multitask learning (MTL) models to capture cross-state variability. We note that some states, such as Massachusetts, have way more accident records available than other states, such as Montana. Thus, combining states with more labels can help predict trends for states with fewer labels. However, to ensure that this pooling strategy works, we also require some structural similarities in the feature representations of different states [60].

Interestingly, we observe cross-sectional trends shared across states. In Figure 2, we visualize the number of accidents across four states, including Delaware, Iowa, Illinois, and Massachusetts. In the top panel, we find that the number of accidents gradually increases until 2019 but dips in 2020 due to lockdowns during the pandemic. Nevertheless, after 2020, the accident count increased back.

Multitask learning [8] encodes such inductive bias by using a shared encoder for all tasks. Let $f(\cdot)$ denote the encoder, e.g., a GNN. For each state, we design a separate prediction layer to map the feature vector to an output. Denote these as $h_1(\cdot), h_2(\cdot), \ldots, h_k(\cdot)$. To train these layers, we combine

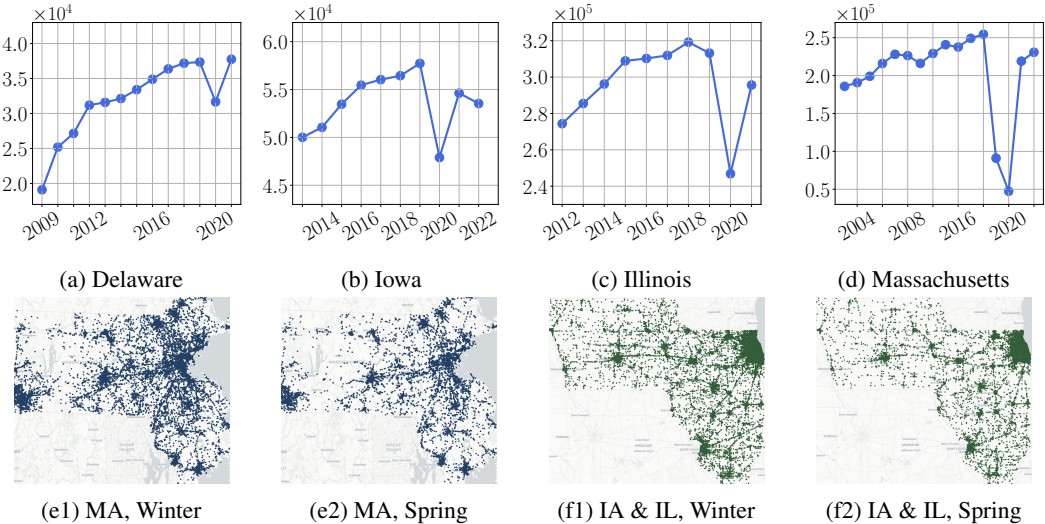

|  |  |  |  |
|---|---|---|---|
| (a) Delaware | (b) Iowa | (c) Illinois | (d) Massachusetts |
| (e1) MA, Winter | (e2) MA, Spring | (f1) IA & IL, Winter | (f2) IA & IL, Spring |

Figure 2: 2a-2d: Showing the evolution of annual accident counts across states. There is a sharp drop in 2020 due to the pandemic. 2e1-2f2: Seasonal pattern of accidents, where more accidents occur during winter compared to spring.

the data from all states and train a multitask model that yields predictions simultaneously for all states. In particular, we minimize the average loss over the combined dataset of all tasks.

**Combining road network geometry and traffic flow.** Lastly, we develop a transfer learning (TL) technique to combine traffic flow information, which has been shown to relate to the crash frequency in regression models [49], with network features. We use annual traffic volume reports to support accident prediction by adding another labeling task to our model. In this task, we use the traffic volume information as edge labels beside the traffic records. Given traffic volume up to time $t$, we aim to predict traffic volume from $t + 1$ onwards. This is a regression task at an annual level: Given an edge and a year, the task is to predict the average traffic volume on the road. Importantly, we exclude the traffic volume feature for this prediction. Then, we combine this task together with accident prediction in a multitask learning model. We can view accident prediction as the primary task $T_1$ and volume prediction as an auxiliary task $T_2$. We jointly train a shared encoder $f(\cdot)$ and two separate prediction layers $h_1(\cdot)$ and $h_2(\cdot)$ on the averaged loss of both tasks.

We plot the distribution of accidents relative to the corresponding daily traffic volume in Figure 3. For volume, we gather the average daily traffic records for each road and aggregate the number of accidents at various intervals. The result indicates a positive correlation between higher traffic volume and increased accidents. Consequently, one might expect a positive transfer from the volume prediction task to the accident prediction task when they are trained jointly in a shared model [35].

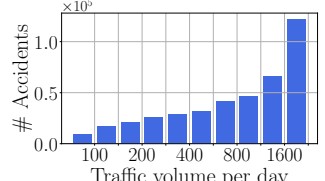

Figure 3: Distribution of accidents by daily traffic volume.

## 3 Experiments

This section evaluates various graph learning methods to predict accident labels based on our new dataset. We focus on three questions. How effective are existing graph learning methods in capturing the patterns of accidents? Does multitask learning capture cross-sectional trends across states in the prediction result? Will traffic volume information help with accident prediction? We provide positive results for the three questions. Additionally, we conduct ablation studies to highlight the importance of graph-structural information and positive correlations at the macro-level as captured by multitask and transfer learning techniques. Lastly, we discuss the implications of our experimental findings.

## 3.1 Experimental setup

**Baselines.** Recall that we consider an edge-level link prediction task. We use both embedding methods and graph neural networks as baselines. First, we test multilayer perceptrons (MLP) using the node features, including node degrees, betweenness centrality, and weather. This tests using node features without network structures. Second, we test embedding methods, including Node2Vec [17] and DeepWalk [48], and add a layer to concatenate the node embeddings and features. Third, we evaluate various GNN architectures, including GCN [34], GraphSAGE [20], and GIN [63]. Lastly, We also evaluate spatiotemporal GNNs that model the temporal dependency of the time serial features. These include DCRNN [39], STGCN [68], AGCRN [1], and Graph Wavenet [62]. To enhance the prediction of GNN, we add Node2Vec embeddings as node features.

We consider three methods to conduct multitask and transfer learning: First, we fit multitask GNNs by combining the data of all eight states. We denote this model as MTL. Second, after fitting an MTL model, we fine-tune the feature encoder on an individual state's data to personalize its representations. We denote this model as MTL-FT. Third, we train a model on accident and volume prediction jointly. We denote this model as TL. For all methods above, we use GraphSAGE as the feature encoder.

**Implementations.** For all baselines, we construct the feature representation for each edge by concatenating the two adjacent nodes' encoded representations and the edge-level features. In our implementation, we fix the dimension of node embeddings as 128. We use two-layer MLP and GNN with a hidden dimensionality of 256. We train our models using Adam as the optimizer. We use a learning rate of 0.001 for 100 epochs on all models. These hyper-parameters are tuned with grid search on the validation set for all the models. The number of layers is tuned in a range of $\{2, 3, 4\}$. The hidden dimensionality is tuned in a range of $\{128, 256, 512\}$. The learning rate is tuned in a range of $\{0.01, 0.001, 0.0001\}$. The number of epochs is tuned in a range of $\{50, 100, 200\}$. See Appendix B for more implementation details.

For each state, we evenly split the available period of accidents into training, validation, and test sets. We split the accidents according to time. We use past accident records until a specific year to train the models and evaluate the model's performance on future accidents occurring after that year. We focus our prediction monthly and add up the daily number of occurrences, but note that one can use our datasets to conduct the analysis at a daily or annual level too. We use both regression and classification metrics to evaluate the results. For regression, we measure the mean absolute error (MAE) between the predicted number of accidents and the actual number of occurrences on a particular road. For classification, we measure AUROC scores.

## 3.2 Experimental results

We summarize our experimental results in Table 2. We highlight three conceptual takeaways below, which we believe will also apply more broadly beyond the specific setting of our experiment.

**(1) Graph neural networks can accurately predict accident labels.** We find that using graph neural networks can predict accident counts with **0.3** mean absolute error, which is **22%** relative to the absolute accident count on average over eight states. For classifying whether an accident occurs or not, GNNs can achieve **87%** AUROC score on average.

Among the GNNs, we observe that the comparisons between GraphSAGE and other spatiotemporal GNNs are generally mixed, with none of them dominating each other. While GraphSAGE has fewer parameters, its performance is still comparable with (if not outperforming) alternative models with two times or more parameters. One explanation is that spatiotemporal GNNs have more trainable parameters than GraphSAGE. Therefore, they need more training labels to fit the model. On the other hand, the labeling rate of our dataset, i.e., the percentage of edges with a positive label or accident occurrence, is around or below 0.2%, as shown in Table 2 (under the row of "Positive Rate").

**(2) Multitask learning captures macro-level trends across states.** We also find that multitask learning outperforms single-task learning (STL) by relatively **8.4%** in terms of MAE and **0.9%** in terms of AUROC averaged over eight states. This is achieved by first training a model on the combined data of all states and then fine-tuning the MTL model on each individual state data.

**(3) Transfer learning with AADT improves test performance.** Lastly, we find that combining traffic volume and accident prediction yields a relative improvement of **7.9%** in MAE and **1.1%** in AUROC over STL, averaged over the four states with traffic volume records.

Table 2: We compare the experimental results across eight states using node embedding methods and graph neural networks. We also include multitask and transfer learning results for each state. We report the results obtained under the mean absolute errors (MAE) on the test split. We also report the AUROC score on the test split. To measure standard deviations, we run the same experiment over three different random seeds and report the averaged result.

| MAE (↓) | DE | IA | IL | MA | MD | MN | MT | NV |
|---|---|---|---|---|---|---|---|---|
| Avg Count | 1.23 | 1.14 | 1.33 | 2.27 | 1.22 | 1.18 | 1.16 | 1.38 |
| MLP | 1.4±0.07 | 0.3±0.02 | 1.4±0.17 | 1.0±0.11 | 0.4±0.02 | 0.3±0.02 | 0.4±0.01 | 0.5±0.01 |
| Node2Vec | 1.1±0.18 | 0.3±0.01 | 0.7±0.05 | 1.3±0.51 | 0.4±0.03 | 0.4±0.02 | 0.2±0.03 | 0.3±0.01 |
| DeepWalk | 0.8±0.05 | 0.3±0.01 | 0.6±0.03 | 1.0±0.06 | 0.4±0.01 | 0.4±0.01 | 0.3±0.03 | 0.3±0.02 |
| GCN | 0.6±0.02 | 0.3±0.02 | 0.5±0.06 | 0.7±0.02 | 0.4±0.04 | 0.3±0.00 | 0.2±0.03 | 0.3±0.02 |
| GraphSAGE | 0.3±0.01 | 0.3±0.01 | 0.4±0.03 | 0.8±0.02 | 0.4±0.01 | 0.3±0.01 | 0.2±0.02 | 0.2±0.01 |
| GIN | 0.8±0.02 | 0.3±0.04 | 0.4±0.04 | 1.1±0.02 | 0.4±0.02 | 0.3±0.06 | 0.2±0.08 | 0.2±0.05 |
| AGCRN | 0.3±0.01 | 0.3±0.01 | 0.4±0.01 | 0.7±0.01 | 0.4±0.01 | 0.3±0.01 | 0.2±0.01 | 0.2±0.04 |
| STGCN | 0.2±0.02 | 0.3±0.00 | 0.4±0.06 | 0.8±0.06 | 0.5±0.01 | 0.3±0.01 | 0.2±0.01 | 0.2±0.01 |
| Graph Wavenet | 0.3±0.03 | 0.3±0.02 | 0.4±0.00 | 0.7±0.01 | 0.3±0.00 | 0.3±0.01 | 0.3±0.03 | 0.2±0.01 |
| DCRNN | 0.3±0.01 | 0.3±0.02 | 0.4±0.03 | 0.9±0.06 | 0.3±0.01 | 0.4±0.02 | 0.2±0.02 | 0.2±0.03 |
| MTL | 0.2±0.02 | 0.2±0.01 | 0.4±0.02 | 0.7±0.02 | 0.3±0.00 | 0.2±0.01 | 0.1±0.00 | 0.2±0.01 |
| MTL-FT | **0.2**±0.01 | **0.2**±0.00 | **0.2**±0.00 | 0.7±0.01 | **0.3**±0.00 | **0.2**±0.01 | **0.1**±0.00 | **0.2**±0.00 |
| TL | 0.2±0.01 | - | - | **0.6**±0.02 | 0.3±0.01 | - | - | 0.2±0.01 |

| AUROC (↑) | DE | IA | IL | MA | MD | MN | MT | NV |
|---|---|---|---|---|---|---|---|---|
| Training Size | 93,184 | 187,046 | 646,739 | 540,682 | 283,226 | 124,435 | 34,475 | 73,164 |
| Positive Rate | 0.23 | 0.07 | 0.14 | 0.10 | 0.15 | 0.05 | 0.05 | 0.12 |
| MLP | 75.5±0.6 | 68.7±0.1 | 71.4±0.3 | 70.6±0.2 | 74.1±0.4 | 76.7±0.2 | 71.6±0.1 | 56.2±0.1 |
| Node2Vec | 83.5±0.1 | 83.8±0.1 | 77.4±1.5 | 70.7±0.4 | 83.5±0.1 | 80.6±0.1 | 84.9±0.1 | 91.8±0.1 |
| DeepWalk | 83.4±0.3 | 81.6±0.2 | 78.6±0.2 | 69.5±0.2 | 83.7±0.1 | 80.5±0.2 | 85.0±0.1 | 91.8±0.1 |
| GCN | 83.2±0.1 | 85.4±0.1 | 84.7±1.0 | 70.6±0.1 | 83.2±0.1 | 84.3±2.2 | 87.4±0.1 | 91.9±0.6 |
| GraphSAGE | 87.6±0.1 | 84.8±0.2 | 87.0±0.2 | 81.8±0.1 | 87.6±0.1 | 83.8±2.8 | 87.5±0.1 | 91.6±1.0 |
| GIN | 82.6±0.7 | 83.5±0.1 | 84.2±1.4 | 68.9±0.5 | 82.6±0.7 | 85.4±1.4 | 85.4±0.1 | 91.4±1.0 |
| AGCRN | 86.0±0.2 | 83.9±0.2 | 86.3±0.3 | 82.1±0.2 | 88.5±0.1 | 81.8±0.7 | 84.3±0.4 | 90.7±0.2 |
| STGCN | 85.4±0.1 | 83.5±0.1 | 85.2±0.4 | 81.9±0.3 | 88.7±0.1 | 81.5±0.2 | 83.8±0.3 | 91.5±0.3 |
| Graph Wavenet | 85.0±0.2 | 83.9±0.2 | 85.8±0.2 | 81.9±0.5 | 87.9±0.1 | 80.3±0.1 | 83.4±0.2 | 90.6±0.2 |
| DCRNN | 81.2±1.2 | 81.8±0.1 | 80.7±0.0 | 70.5±0.1 | 84.5±0.3 | 79.3±0.4 | 81.9±0.5 | 90.5±0.7 |
| MTL | 87.7±0.1 | 81.7±0.2 | 84.4±0.3 | 79.6±0.1 | **88.7**±0.1 | **87.9**±0.0 | 88.4±0.2 | 90.3±0.2 |
| MTL-FT | **87.8**±0.3 | **84.9**±0.2 | **87.2**±0.2 | 81.9±0.3 | 88.1±0.1 | 87.6±0.3 | **88.5**±0.3 | 91.8±0.2 |
| TL | 87.3±0.2 | - | - | **82.6**±0.2 | 87.9±0.4 | - | - | **92.8**±0.1 |

## 3.3 Ablation studies

**Influence of graph-structural features.** We conduct a leave-one-out analysis of different categories of features for accident prediction. These include graph-structural features, weather, and traffic volume. We remove one type of feature at a time and compare the performance after leaving out a particular feature. Removing graph-structural features reduces the performance by 6.9%. On the other hand, removing weather and traffic volume reduces performance by 2.3% and 1.2%, respectively. See Table 3 for the results, which justifies graph-structural features are the most significant features.

**Positive pairwise transfer across states.** Next, we measure the transfer effects between every pair of states. For each state, we view it as a source state and consider how well it would transfer to another target state. To test this, we conduct MTL by combining each state with another state's data. This leads to a total of 28 pairwise MTL models. We show the results in Figure 4 on the right. To help read this table, we subtract each MTL model's performance for a target state from the STL performance of that target state. Thus, a positive value in the table indicates a positive transfer from the source state to the target state. We find that for most pairs, the effect is positive.

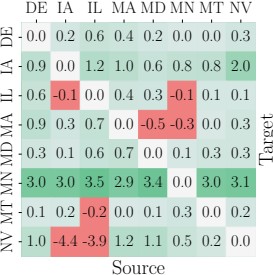

Figure 4: Pairwise MTL vs STL.

**Transferability from traffic volume to accident prediction.** We also measure the transfer effect from volume prediction to accident prediction. For one state, such as the MA, we first fit a model on the volume prediction task. Then, we fine-tune this model on the accident prediction task using

Table 3: We evaluate the influence of graph-structural structure, weather, and traffic volume information. We report the test AUROC of accident classification by removing each feature type in the network. We also include the transfer results from traffic volume to accident prediction.

| | DE | MA | MD | NV |
|---|---|---|---|---|
| Using all features | 87.61±0.10 | 81.80±0.12 | 87.51±0.02 | 91.62±0.99 |
| w/o graph structural features | 81.25±0.51 | 79.63±0.18 | 79.77±0.94 | 82.56±0.09 |
| w/o weather information | 87.27±0.32 | 80.71±0.26 | 80.22±0.45 | 90.38±0.88 |
| w/o road information | 82.99±0.54 | 81.65±0.77 | 80.63±0.52 | 84.24±0.41 |
| w/o traffic volume information | 87.15±0.49 | 80.94±0.31 | 86.17±1.15 | 91.58±0.99 |
| Combining traffic volume prediction | 87.78±0.07 | 82.18±0.14 | 87.77±0.24 | 92.07±0.57 |

the accident labels while keeping the same network and other features. This fined-tuned model outperforms single-task learning (STL) by 0.6%, averaging over four states with volume labels.

**Sensitivity analysis.** Lastly, we study the hyper-parameters used in our experiments. We set the hyper-parameters as follows: the number of layers is 2, the hidden dimensionality is 256, the learning rate is $1e^{-3}$, and the number of epochs is 100. We then vary one hyper-parameter at a time and keep the others unchanged. We notice that using the number of layers as 2, hidden dimensionality as 256, and learning rate as $1e^{-3}$ yields the best results for all baselines. The validation performance stops improving after training up to 100 epochs. Thus, we adopt these settings as the default parameters.

### 3.4 Interpretations and implications

Our findings provide some evidence to show that the road network structure is highly predictive of traffic accident occurrences. The evaluation of numerous graph learning methods shows that these methods are valid for accident analysis. Our findings about the road structures go beyond existing regression analysis in the transportation literature (e.g., [49, 6, 41]) thanks to our large-scale dataset. Based on the findings, we discuss several implications for when our datasets might be helpful.

*Studying the effect of policy interventions:* One way to use our dataset is for policy interventions by examining the accident patterns before and after the implementation of this policy. Our modeling approach could be useful in policy making such as the abolition of mandatory vehicle inspection. The model predictions can provide counterfactual comparisons to facilitate such discussions.

*Variability at the county level:* With our dataset, it is also possible to study road structures across different counties within the same state. This controls for variability with weather conditions, and we believe the comparison can potentially inform network design as well.

*Better reporting of volume information:* We believe that better reporting of traffic volume information could be very useful for accident analysis. In our dataset, we notice that the percentage of roads where volume reports are available is pretty low (cf. Table 1). For example, we noticed that on some roads the accident numbers are high but the traffic volume is either low or missing. Besides, we also note that it is likely there is under-reporting of accidents (e.g., to reduce insurance costs). Thus, having better reports of traffic flow combined with better modeling can help address such questions.

## 4 Related Work

**Graph datasets and benchmarks.** Datasets and benchmarks are instrumental to machine learning research. Motivated by developments in machine learning on graphs, several large-scale graph learning benchmarks have been developed [26, 45, 14, 25]. Chiang et al. [10] construct a large-scale graph dataset corresponding to Amazon's product co-purchase information. MoleculeNet [61] introduces a large-scale benchmark for studying molecule graphs and structures. See also recent works by Townshend et al. [55], who study three-dimensional representations of molecules, and Paetzold et al. [47], who provide whole-brain vessel graphs based on whole, segmented murine brain images. TAPE [51] describes a set of five biologically relevant semi-supervised learning tasks spread across different domains of protein biology. TUDataset [45] consists of over 120 datasets of varying graph sizes from various applications.

Table 4: Comparison of our dataset to the work of Huang et al. [27] and Yuan et al. [70].

|  | Huang et al. [27] | Yuan et al. [70] | Our Dataset |
| --- | --- | --- | --- |
| Data Source | Based on work by Moosavi et al. (2019), sourced from Microsoft Bing Map Traffic | Department of Transportation's official accident reports | Department of Transportation's official accident reports |
| Coverage | 2.8 million records across 13 states for 2016-2021 | Iowa, for 2006-2013 | 9 million records, across 8 states, for a maximum of 20 years up to 2023 |
| Prediction Tasks | Node-level classification, evaluated on AUROC | Spatial grid-level regression, evaluated under MAE | Edge-level regression/classification, evaluated under MAE/AUROC |
| Features | Road network features | Averaged grid-level features, traffic volume, rainfall features | Traffic volume, weather features, road network features |

The open graph benchmark and database [26, 14, 25] involve academic collaboration networks, protein structural graphs, Reddit social networks, and carefully designed data splits to facilitate comparison between methods. Besides supervised learning, recent work has developed benchmarks for graph contrastive learning [75], continual graph learning [72], node outliner detection [42], and anomaly detection [28]. Compared with existing datasets, ours concerns a problem that is of relevance to society, and we provide spatiotemporal patterns across states.

**Spatiotemporal graph mining.** The importance of spatial and temporal features for time series prediction is well recognized [7, 58, 40]. Convolutional networks are often used to process time series graphs [68, 59]. Besides, multitask and meta-learning is used to tackle spatiotemporal heterogeneity [38, 65, 57]. Zhuang et al. [76] quantify the uncertainty of spatiotemporal GNN. Lastly, the transferability of graph representations has recently been studied by Zhu et al. [74] for airport networks and gene interaction graphs. Our results suggest that road network structures are highly transferable in predicting traffic accidents. The use of contrastive learning on graph-structured data has been extensively explored in recent work [67, 66]. The following studies [50, 71] have formulated contrastive learning methods on spatiotemporal graph data, such as urban flow prediction, crime prediction, and house price prediction. One interesting question is to revisit these techniques within the context of traffic accident prediction for road safety.

**Data-driven traffic forecasting.** There is a vast literature on traffic network analysis such as traffic forecasting [31]. Public datasets such as METR-LA and PEMS-BAY are widely used in ML research. These are collected from loop detectors by California's Transportation Agencies. Many research studies use taxi data, such as the NYC open dataset and Didi Chuxing traffic information. Recent work has studied constructing road networks from satellite images [22]. Some recent studies of traffic forecasting are driven by ride-hailing platforms [16]. There are also studies on predicting the estimated time of arrival [24], traffic time and distance for a taxi trip [38], and traffic speed [21].

**Comparison with traffic accident prediction datasets.** Previous works [70, 27] and our work are targeted at large-scale traffic accident prediction. Similar to our work, both works use OpenStreetMaps to construct road network features. Besides, our work and previous works include similar road network features, including the length of a road and the type of road (e.g., highway or residential, one-way).

The difference between our work and previous works is primarily in constructing the accident information. First, TAP [27] used the accident information collected from another work by Moosavi et al. [44], sourced from Microsoft Bing Map Traffic. Their dataset includes a total of 2.8 million accident records, which is for five years (2016-2021). In contrast, our datasets are collected from the Department of Transportation's official accident reports. Our dataset includes 9 million records for a maximum of 20 years (e.g., Massassuchetts, from 2002 to 2023, which also contains more recent data). Second, each accident record contains the (latitude and longitude) of the incident. We map each coordinate to the nearest edge, leading to an edge-level classification/regression problem, whereas TAP maps each record to the nearest node, resulting in a node-level classification setting. Thus, our work can be evaluated under both MAE/AUROC metrics. Another major difference is that we have collected traffic volume and weather features alongside the road network features. In summary, we give a detailed comparison between our dataset and two existing datasets in Table 4.

Table 5: Comparison of our dataset to several existing spatiotemporal datasets.

| | METR-LA [39] | PEMS-Bay [39] | Taxi NYC [54] | Didi Chuxing [38] | Our Dataset |
|---|---|---|---|---|---|
| Targets | Traffic speed and volume | Traffic speed and volume | # Rides in a region | Trip time | Traffic accident |
| Coverage | 6 million for 4 months from Mar. 2012 to Jun. 2012 | 16 million for 6 months: Jan. 2017 - May 2017 | 35 million over 5 years: Jan. 2011 - Jun. 2016 | 61.4 million over six months: May. 2017 - Oct. 2017 | 9 million in 8 states up to 20 years until 2023 |
| Data Source | Highway loop detectors | CalTrans | Taxi GPS data for NYC | Didi Chuxing Beijing | Department of Transportation |
| Features | Traffic speed and volume | Traffic speed and volume | Traffic flow, latitude, longitude, distances | Travel distance, # road, lights, turns for each ride | Traffic volume, weather, and road network features |

**Comparison with spatiotemporal datasets.** Next, we compare several spatiotemporal datasets with ours. We note that existing datasets focus on rather different tasks. Li et al. [39], proposes to predict traffic speed and volume collected from highway loop detections. Some other works study to predict traffic information from ride-hailing platforms, such as the ride number in a certain region [54] and the estimated trip time [38]. In contrast, our work aims to predict traffic accidents. In addition, our dataset provides extensive coverage over 8 states. Other spatiotemporal datasets are mostly constructed for a single city. In contrast, our dataset covers a broader range of areas and facilitates the study of cross-sectional trends beyond a single area. For details, see Table 5 below.

## 5 Discussions

**Interpretations and future directions.** It is worth pointing out that our goal is not to identify exactly where new accidents will occur since this is impossible. Instead, our results can be interpreted as providing suggestions for the risk or hazard of a particular road, whether or not an accident will occur [23]. We can use the predicted accident count as a risk oracle for guiding drivers' behavior [2]. With this in mind, we discuss several promising questions for future work. First, one can consider the accident prediction problem with our dataset and methodology but at different granularities. One can capture variability within a week by predicting at a daily level or even an hourly level [73]. This can be used to compare risks between weekdays vs. weekends and holidays or rush hour vs. evening hours. Second, one can develop novel time series graph learning methods such as autoregressive models [29]. Besides predicting crash frequency, another relevant statistic to consider is crash rate as normalized by vehicle millage. Third, it would be interesting to see if recent developments in road network construction from satellite images can be used to design new predictive features. More broadly, there are ample opportunities to use our datasets to study policy interventions on road safety.

**The ML4RoadSafety package.** To facilitate further research, we have developed a package to make our datasets easily accessible to researchers. Our package uses the data format as the existing graph learning library, PYTORCH GEOMETRIC, fully compatible with PYTORCH. A user can have access to our dataset with a single line of code, by only specifying the name of a state, such as Massachusetts. The package will automatically download, store, and return a dataset object. Then, our package offers functions to obtain the accident records and network features for a particular month. Besides the datasets, we provide a trainer module to train and evaluate a GNN on our datasets. Using the trainer, the user can easily implement other training techniques, including multitask learning, transfer learning, and contrastive learning. We provide code examples in Appendix B.3.

## 6 Conclusion

We collected a large-scale dataset of 9 million traffic accident records across eight states to analyze traffic accident occurrences using road networks, traffic flow, and weather reports. Through extensive experiments, we found that existing graph neural networks can be used to predict accident labels with over 87% AUROC score. This uses multitask and transfer learning techniques on graphs. Our analysis reveals strong cross-sectional similarities across states regarding road network structures. Ablation studies validate the importance of graph-structural features for achieving the results.

**Acknowledgement.** Thanks to Hari Balakrishnan for an invited talk at Northeastern that brought the problem [2] to our attention. Thanks to Ravi Sundaram for several discussions. This research is supported in part by Northeastern University's Transforming Interdisciplinary Experiential Research (TIER) 1: Seed Grant/Proof of Concept Program.

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

# A   Data Collection Procedure

In this section, we describe the details of the collection of our dataset. We have provided the implementation in our code repository: `https://github.com/VirtuosoResearch/ML4RoadSafety`. We have also uploaded the entire dataset to an open repository: `https://doi.org/10.7910/DVN/V71K5R`. This section serves as the documentation for the collection process. For reference, we have collected the links to public data sources where we extracted our dataset below.

Table 6: Links to the data sources from which we extracted our dataset.

| Traffic accident records | |
| --- | --- |
| Delaware Open Data | `https://data.delaware.gov/Transportation/Public-Crash-Data-Map/3rrv-8pfj` |
| Delaware DOT | `https://deldot.gov/search/` |
| Iowa DOT | `https://icat.iowadot.gov/` |
| Illinois DOT | `https://gis-idot.opendata.arcgis.com/search?collection=Dataset&q=Crashes` |
| Mass DOT | `https://apps.impact.dot.state.ma.us/cdp/home` |
| Maryland Open Data | `https://opendata.maryland.gov/Public-Safety/Maryland-Statewide-Vehicle-Crashes/65du-s3qu` |
| MN Crash | `https://mncrash.state.mn.us/Pages/AdHocSearch.aspx` |
| Montana DOT | `https://www.mdt.mt.gov/publications/datastats/crashdata.aspx` |
| Nevada DOT | `https://ndot.maps.arcgis.com/apps/webappviewer/index.html?id=00d23dc547eb4382bef9beabe07eaefd` |

| Road networks | |
| --- | --- |
| OSMnx Street Network Dataverse | `https://dataverse.harvard.edu/dataset.xhtml?persistentId=doi:10.7910/DVN/CUWWYJ` |
| OpenStreetMap Road Categories | `https://wiki.openstreetmap.org/wiki/Highways#Classification` |
| Google Map API | `https://maps.google.com/` |

| Weather reports | |
| --- | --- |
| Meteostat API | `https://meteostat.net/en/` |

| Traffic volume reports | |
| --- | --- |
| Delaware | `https://deldot.gov/search/` |
| Maryland | `https://data-maryland.opendata.arcgis.com/datasets/mdot-sha-annual-average-daily-traffic-aadt-segments/explore?location=38.833256%2C-77.269751%2C8.30&showTable=true` |
| Massasuchetts | `https://mhd.public.ms2soft.com/tcds/tsearch.asp?loc=Mhd&mod=` `https://www.mass.gov/lists/massdot-historical-traffic-volume-data` |
| Nevada | `https://geohub-ndot.hub.arcgis.com/datasets/trina-stations/explore?location=38.490765%2C-116.969086%2C7.27&showTable=true` |

## A.1   Traffic accident records

First, we construct the accident labels in our dataset. We collect accident records for each state from the state's Department of Transportation website. Each accident is associated with a report detailing the date and coordinates (i.e., latitude and longitude) of the accident. We collect accident data from eight states. The records are available in the sources listed in Table 6. Then, we map each accident to the closest road in the road network according to the coordinates of the accident. Specifically, we use the accident's location, denoted as $c$, and compare it with the coordinates of an edge's two endpoints, labeled as $a$ and $b$. We map the accident to the edge with the smallest value of $D(a, b) - (D(a, c) + D(b, c))$ where $D(\cdot)$ is the Euclidean distance metric.

## A.2 Road networks

We generate a road network for a state as a directed graph, based on the road network structure in the state extracted from OpenStreetMap. Nodes are defined as all the publicly available intersections on the roads in OpenStreetMap, and edges are the road segments between these nodes. Therefore, one road can have multiple edges depending on the number of intersections with other roads. There would be three edges if there are three intersections on a road.

The derived edges in the graph include road segments of various levels in a state, including every city, town, urbanized area, county, census tract, and Zillow neighborhood in the state. We obtain the above information from the OSMnx Street Networks Dataverse in OpenStreetMap (cf. Table 6).

## A.3 Road network features

We describe four types of features associated with road networks in the following.

**Graph structural features (Node-level, static).** All nodes in a graph are associated with static structural features, including node in-degrees, out-degrees, and betweenness centrality scores. For node degrees, we encode node degree values as one-hot vectors with a dimension of the graph's max degree. We encode two vectors for every node's in-degree and out-degree. For betweenness centrality scores, we generate a real-value feature that measures the ratio of shortest paths between all node pairs that contain a certain node.

**Historical weather information (Node-level, temporal).** Besides, each node is also associated with daily weather information. We collect the following six features related to the weather conditions within a particular month: (i-iii) the maximum, minimum, and average of the temperature on the road's surface; (iv) the total precipitation, including rainfall or snowfall; (v) the average wind speed; (vi) the sea level air pressure. For every node, All the weather conditions are measured at the nearest meteorological station to the node according to its geometric coordinate. The weather information is extracted using Meteostat API (cf. Table 6). Since this data is temporal in nature and is available at every node, we believe that the weather features would be important for our task.

**Road information (Edge-level, static).** All edges are associated with static features describing length information and road categories. We encode the physical length of the road in meters as a real-value feature. We encode the road category as a 24-dimensional characteristic vector for each edge, with each entry as either 0 or 1, indicating whether the road is associated with the category. Each road can be assigned (multiple) categories from 24 road categories. These include a oneway, primary, primary link, secondary, secondary link, access ramp, bus stop, crossroad, disused, elevator, escape, living street, motorway, motorway link, residential, rest area, road, stairs, tertiary, tertiary link, trunk, trunk link, unsurfaced, and unclassified. We obtain the road information from OpenStreetMap.

**Traffic volume (Edge-level, temporal).** Each edge is also associated with a traffic volume feature that is measured yearly. We encode the traffic volume as a real-value feature that measures the average number of cars traveled on the road over a year. We extract the information from the Annual Average Daily Traffic reports published by the Department of Transportation of each state. These reports provide information on the traffic volume for a sample of streets within the state. By using the Google Maps API, we extract the corresponding coordinates for the street names and map them to the edges of the road network.

## A.4 Summary of road network features

Here is a list of node-level features we have included in our dataset:

- Latitude,
- Longitude,
- Node indegree and outdegree,
- Betweenness centrality,
- Average surface temperature (tavg),
- Max surface temperature (tmax),
- Min surface temperature (tmin),

- Total precipitation (prcp),
- Avg wind speed (wspd),
- Sea level air pressure (pres).

Here are the edge-level features in our dataset:

- A binary label that indicates whether the road is one way or not,
- A multi-class label that indicates whether the road is highway, residential, etc.,
- Length of the road,
- Annual average daily traffic (AADT), if this information is available in the report.

# B  Experiment Details

In this section, we describe the details of our experiments that were left out in the main text. First, we describe additional implementation details. Then, we describe the omitted experimental results. These include the ablation study of graph structural features, transferability from traffic volume to accident prediction, various metrics for evaluating accident prediction, results of hyper-parameter tuning, preliminary results of contrastive learning, and observations of accident counts across seasons. Lastly, we provide examples of using our package.

## B.1  Implementation details

In our implementation, we set the dimension of node embeddings as 128, the number of layers as 2, and the hidden dimensionality as 256. We train our models using Adam as the optimizer. We use a learning rate of 0.001 for 100 epochs on all models. The hyper-parameters are determined by searching in the following ranges: The hidden dimensionality is tuned in a range of $\{128, 256, 512\}$. The number of layers is tuned in a range of $\{2, 3, 4\}$. The learning rate is tuned in a range of $\{0.01, 0.001, 0.0001\}$.

For each state, we evenly split the available period of accidents into training, validation, and test set. Table 7 summarizes the dataset splitting for each state. While our evaluation focuses on monthly predictions, our datasets can also be utilized for conducting analyses at daily or annual levels.

Table 7: Data splitting of accident records of eight states.

|      | Train (years) | Train (records) | Valid (years) | Valid (records) | Test (years) | Test (records) |
|------|---------------|-----------------|---------------|-----------------|--------------|----------------|
| DE   | 2009 - 2012   | 112,670         | 2013 - 2017   | 174,278         | 2018 - 2022  | 171,311        |
| IA   | 2013 - 2016   | 213,019         | 2017 - 2019   | 171,455         | 2020 - 2022  | 156,065        |
| IL   | 2012 - 2014   | 856,057         | 2015 - 2017   | 949,745         | 2018 - 2021  | 1,174,900      |
| MA   | 2002 - 2008   | 1,265,895       | 2009 - 2015   | 933,786         | 2016 - 2022  | 1,096,885      |
| MD   | 2015 - 2017   | 341,902         | 2018 - 2019   | 229,446         | 2020 - 2022  | 306,995        |
| MN   | 2015 - 2017   | 148,361         | 2018 - 2019   | 154,150         | 2020 - 2022  | 188,858        |
| MT   | 2016 - 2017   | 40,040          | 2018          | 20,677          | 2019 - 2020  | 39,222         |
| NV   | 2016 - 2017   | 101,975         | 2018          | 48,854          | 2019 - 2020  | 86,509         |

**Number of parameters in each model.** This detail complements our discussion in Section 3.2:

- MLP, Node2Vec, DeepWalk: 75K
- GCN, GraphSAGE: 253K
- GIN: 778K
- AGCRN: 294K
- STGCN: 713K
- Graph Wavenet: 1,154K
- DCRNN: 1,463K

Table 8: We report the precision and recall scores on the test split on eight states, using node embedding methods, graph neural networks, and multitask and transfer learning methods. We run the experiment over three different random seeds and report the averaged result and standard deviations.

| Precision | DE | IA | IL | MA | MD | MN | MT | NV |
|---|---|---|---|---|---|---|---|---|
| Training Size | 93,184 | 187,046 | 646,739 | 540,682 | 283,226 | 124,435 | 34,475 | 73,164 |
| Positive Rate | 0.23 | 0.07 | 0.14 | 0.10 | 0.15 | 0.05 | 0.05 | 0.12 |
| MLP | 4.99±0.1 | 1.78±0.0 | 2.54±0.2 | 3.52±0.6 | 3.47±0.1 | 1.87±0.0 | 0.87±0.0 | 3.30±0.0 |
| Node2Vec | 12.76±0.2 | 3.16±0.3 | 3.52±0.5 | 3.28±0.8 | 5.64±0.4 | 2.38±0.3 | 3.50±0.2 | 10.74±0.0 |
| DeepWalk | 14.13±0.5 | 2.90±0.5 | 3.37±0.6 | 3.30±0.1 | 4.88±0.1 | 2.64±0.3 | 2.62±0.4 | 7.74±0.0 |
| GCN | 11.09±0.7 | 2.36±0.1 | 7.54±0.7 | 4.05±0.4 | 5.60±0.1 | 4.60±0.1 | 5.27±0.2 | 9.17±0.1 |
| GraphSAGE | 18.56±0.9 | 4.13±0.2 | 8.54±0.3 | 4.71±0.2 | 7.04±0.1 | 6.26±0.4 | 4.11±0.5 | 8.62±0.0 |
| GIN | 13.95±0.6 | 3.63±0.2 | 9.84±0.8 | 4.45±0.8 | 6.50±0.5 | 4.08±0.7 | 5.72±0.7 | 10.27±0.0 |
| AGCRN | 11.36±0.6 | 3.50±0.2 | 7.60±1.1 | 4.30±0.3 | 6.61±0.2 | 5.66±0.2 | 3.12±0.1 | 7.74±0.1 |
| STGCN | 12.33±0.2 | 5.20±0.2 | 7.14±1.7 | 4.54±0.9 | 8.91±0.1 | 4.65±0.1 | 4.04±0.9 | 9.72±0.9 |
| Graph Wavenet | 15.56±0.5 | 3.83±0.3 | 7.76±0.7 | 4.22±1.6 | 7.21±0.3 | 6.62±0.5 | 3.36±0.1 | 7.84±0.1 |
| DCRNN | 11.33±0.5 | 3.87±0.4 | 6.16±0.1 | 4.67±0.1 | 4.75±0.4 | 3.62±0.1 | 3.66±0.4 | 10.06±0.1 |
| STL w/ GraphSAGE | 18.56±0.9 | 4.13±0.2 | 8.54±0.3 | 4.71±0.2 | 7.04±0.1 | 6.26±0.4 | 4.11±0.5 | 8.62±0.0 |
| MTL w/ GraphSAGE | 14.34±0.1 | 3.52±0.4 | **13.62±0.7** | 5.11±0.1 | 8.66±0.0 | 4.44±0.2 | **5.85±0.3** | 16.80±0.0 |
| MTL-FT w/ GraphSAGE | **18.61±0.1** | **4.66±0.0** | 13.61±0.5 | **5.20±0.1** | **8.76±0.1** | **6.66±0.3** | 5.72±0.0 | **16.85±0.4** |
| TL w/ GraphSAGE | 14.10±0.4 | - | - | 5.07±0.4 | 8.51±0.4 | - | - | 9.5±0.1 |

| Recall | DE | IA | IL | MA | MD | MN | MT | NV |
|---|---|---|---|---|---|---|---|---|
| MLP | 60.4±3.7 | 83.8±2.8 | 81.1±2.5 | 79.9±1.4 | 67.2±2.0 | 70.3±1.2 | 72.0±1.6 | 74.3±0.4 |
| Node2Vec | **83.8±1.3** | **89.5±1.6** | 78.2±0.6 | 80.1±3.1 | 80.0±2.3 | **80.0±2.2** | 73.6±0.9 | 83.9±0.5 |
| DeepWalk | 83.2±2.9 | 80.7±4.8 | 81.0±1.1 | 85.7±3.1 | **86.2±3.6** | 78.4±2.9 | 74.0±3.3 | **88.9±0.0** |
| GCN | 75.2±3.0 | 74.0±2.0 | **84.1±0.8** | 82.5±2.1 | 79.1±2.6 | 70.7±1.8 | 63.6±3.8 | 85.4±0.8 |
| GraphSAGE | 60.9±2.7 | 58.5±2.0 | 72.7±2.4 | 51.0±1.3 | 68.7±1.1 | 54.5±2.2 | 59.6±2.9 | 84.7±1.1 |
| GIN | 65.8±3.7 | 75.0±2.4 | 78.0±4.3 | 48.2±2.0 | 78.3±3.2 | 74.8±2.4 | 65.6±1.8 | 88.6±0.9 |
| AGCRN | 71.5±1.6 | 71.7±1.9 | 73.4±1.0 | 58.2±1.6 | 75.4±1.5 | 73.1±1.5 | 70.7±1.3 | 82.6±3.1 |
| STGCN | 82.9±0.2 | 78.2±4.7 | 75.3±1.2 | 60.7±1.3 | 77.6±1.0 | 75.8±0.3 | 72.6±1.1 | 83.0±0.0 |
| Graph Wavenet | 73.5±3.0 | 67.7±0.7 | 78.3±0.2 | 77.7±1.1 | 73.0±0.2 | 72.9±3.2 | 66.1±1.0 | 79.3±0.1 |
| DCRNN | 75.4±2.6 | 71.1±2.2 | 80.5±1.7 | 81.5±2.0 | 83.0±1.4 | 63.6±1.3 | 66.3±1.0 | 83.4±0.8 |
| STL w/ GraphSAGE | 60.9±2.7 | 58.5±2.0 | 72.7±2.4 | 51.0±1.3 | 68.7±1.1 | 54.5±2.2 | 59.6±2.9 | 84.7±1.1 |
| MTL w/ GraphSAGE | 63.8±1.8 | 78.6±1.0 | 75.6±1.3 | 75.5±0.9 | 78.1±1.2 | 77.6±0.7 | 74.2±0.9 | 82.3±0.5 |
| MTL-FT w/ GraphSAGE | 64.5±0.7 | 78.2±2.5 | 76.7±1.2 | 73.7±1.1 | 77.4±2.1 | 73.2±0.8 | **74.3±3.6** | 84.9±1.2 |
| TL w/ GraphSAGE | 66.7±1.4 | - | - | **92.7±1.3** | 81.7±1.6 | - | - | 84.1±3.3 |

Table 9: Ablation study of different hyper-parameters, including the number of layers, the hidden dimensionality, the learning rate, and training epochs. We report the AUROC scores on the validation split on the Delaware (DE) state dataset using GraphSAGE and DCRNN. We run the same experiment over three random seeds and report the average result.

| | GraphSAGE | | | DCRNN | | |
|---|---|---|---|---|---|---|
| Number of layers | 2 | 3 | 4 | 2 | 3 | 4 |
| | **85.2±0.1** | 84.9±0.3 | 84.4±0.4 | **67.8±1.2** | 67.2±0.8 | 67.3±0.5 |
| Hidden dimensionality | 128 | 256 | 512 | 128 | 256 | 512 |
| | 84.5±0.4 | **85.2±0.1** | 84.5±0.5 | 66.9±0.7 | **67.8±1.2** | 66.9±1.1 |
| Learning rate | $1e^{-2}$ | $1e^{-3}$ | $1e^{-4}$ | $1e^{-2}$ | $1e^{-3}$ | $1e^{-4}$ |
| | 85.0±0.7 | **85.2±0.1** | 84.0±0.5 | 66.8±1.0 | **67.8±1.2** | 66.5±0.9 |
| Epochs | 50 | 100 | 200 | 50 | 100 | 200 |
| | 84.0±0.2 | **85.2±0.1** | **85.2±0.3** | 66.4±0.7 | **67.8±1.2** | **67.8±1.0** |

## B.2 Additional experimental results

**Tranferability from traffic volume to accident prediction.** Next, we report the results of transferring a model from traffic volume to accident prediction. For one state, we first fit a model on the volume prediction task and then fine-tune the model on the accident prediction task. Table 3 reports the performance of the fine-tuned model on accident prediction tasks for four states with traffic volume information. The fine-tuned model outperforms STL by 0.6% on average over the four states.

Table 10: We report the results of applying graph contrastive learning on our datasets using Graph-SAGE and DCRNN. We report the AUROC scores on the test split across four states. We run the same experiment over three random seeds and report the average result.

|  | DE | MA | MD | NV |
|---|---|---|---|---|
| GraphSAGE | 87.6±0.1 | 81.8±0.1 | 87.5±0.0 | 91.6±0.9 |
| GraphSAGE w/ GCL | 86.7±0.2 | 82.4±0.8 | 85.9±0.4 | 91.8±0.4 |
| DCRNN | 81.2±1.2 | 70.5±0.1 | 84.5±0.3 | 90.5±0.7 |
| DCRNN w/ GCL | 86.6±0.3 | 82.5±0.7 | 87.8±0.9 | 91.7±0.4 |
| STGCN | 85.4±0.1 | 81.9±0.3 | 88.7±0.1 | 91.5±0.3 |
| STGCN w/ GCL | 86.0±0.1 | 81.7±0.1 | 89.7±0.5 | 92.4±0.1 |

**Detailed results of classification metrics.** Next, we report the detailed results of classifying whether an accident occurred on a particular road. Table 8 reports the recall and precision scores of the predictions. First, we observe that for all baselines, the recall scores are higher than the precision scores. Using the graph neural networks can predict whether an accident occurred on a road with 10% precision and 85% recall on average over eight states. Second, we also observe that multitask learning outperforms STL relatively by 20% and 21% in terms of precision and precision, respectively. Third, combining traffic volume with accident prediction also improves the STL relatively by 4% and 15% in terms of precision and recall scores.

We observe that while using MLP on node embeddings achieves higher recall than graph neural networks, graph neural networks achieve higher precision scores. This indicates that MLP models tend to be more over-confident when classifying the likelihood of an accident occurring on a road. Given the low precision score, we report the AUROC score in the main text, which provides a summary of the recall score across all decision thresholds.

**Detailed results of hyper-parameter tuning.** We ablate the common hyper-parameters used in our experiments, including the number of layers, the hidden dimensionality, the learning rate, and the number of epochs. In each ablation study, we vary one hyper-parameter and keep the others unchanged. The fixed hyper-parameters are used as follows: the number of layers of 2, a hidden dimensionality of 256, a learning rate of $1e^{-3}$, and 100 epochs.

Table 9 shows the validation AUROC scores, varying hyper-parameters for GraphSAGE and DCRNN on the Delaware state dataset. We notice that using the number of layers as 2, hidden dimensionality as 256, and learning rate as $1e^{-3}$ yields the best results for both baselines. The validation performance stops improving after training the model up to 100 epochs. We also find that these hyper-parameters are useful for other models. Thus, we adopt these settings as the default parameters in the experiments.

**Applying graph contrastive learning.** We conduct a preliminary study of applying graph contrastive learning using GraphSAGE and DCRNN as the base model across four states. We compare them with supervised learning of the baselines. Table 10 shows the results. We find that graph contrastive learning can improve the test performance of the baselines in some states.

**Seasonal patterns of road accidents.** We study how the number of accidents evolves within a year and explore its potential association with seasonal patterns. We hypothesize that the accidents may be affected by the weather and show seasonal trends. To examine this, we aggregate accident counts within four seasons in a year for each state. Specifically, we aggregate accidents occurring from December to February for winter, from March to May for spring, from June to August for summer, and from September to November for fall. Figure 2 shows trends of accident numbers across the seasons for the four states. We notice a significant disparity in accident counts between winter and fall compared to spring and summer. The disparity suggests that accidents may indeed be influenced by seasonal factors, including severe weather conditions and road hazards that are more prevalent during the colder months.

## B.3 Examples for using the ML4RoadSafety package

We provide examples for accessing the data and training models using our dataset. Our package uses the same data format as the existing graph learning library, i.e., PYTORCH GEOMETRIC, which is

Figure 5: Seasonal trend of accident counts within a year. We observe higher accident counts during Winter and Fall compared to Spring and Summer.

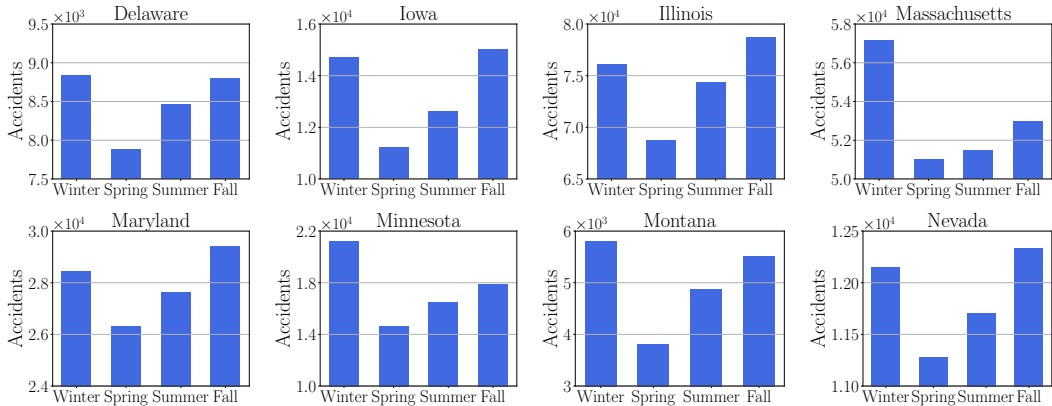

fully compatible with PYTORCH. As shown in Code Snippet 1, our package provides access to our dataset with a single line of code, where the user only needs to specify the name of a state, such as Massachusetts. The package will automatically download, store, and return the dataset object. Then, the user can use a function to obtain the accident records and network features for a particular month.

```
>>> from ml_for_road_safety import TrafficAccidentDataset
# Creating the dataset as PyTorch Geometric dataset object
>>> dataset = TrafficAccidentDataset(state_name = "MA")
# Loading the accident records and traffic network features of a particular month
>>> data = dataset.load_monthly_data(year = 2022, month = 1)
# Pytorch Tensors storing the list of edges with accidents and accident numbers
>>> accidents, accident_counts = data["accidents"], data["accident_counts"]
# Pytorch Tensors of node features, edge list, and edge features
>>> x, edge_index, edge_attr = data["x"], data["edge_index"], data["edge_attr"]
```

Code Snippet 1: ML4RoadSafety Data Loader

Our package provides a trainer class that implements training a graph neural network on one state in our dataset. The trainer class contains the logic for both training and evaluation. As shown in Code Snippet 2, the user can create a trainer object by specifying a model, a dataset, and a corresponding evaluation metric. Then, the user can use a function to launch the training and obtain evaluation results after the training process.

```
>>> from ml_for_road_safety import Trainer, Evaluator, TrafficAccidentDataset
# Creating the dataset
>>> dataset = TrafficAccidentDataset(state_name = "MA")
# Get an evaluator for accident prediction, e.g., the classification task.
>>> evaluator = Evaluator(type = "classification")
# Initialize a trainer with a GNN model, a dataset, and an evaluator
>>> trainer = Trainer(model, dataset, evaluator, ...)
# Conduct training and evaluation inside the trainer
>>> log = trainer.train()
```

Code Snippet 2: ML4RoadSafety Trainer

### B.3.1   Implementation of multitask and transfer learning

In multitask learning, we combine multiple datasets and optimize the average loss of the combined data. We implement this as follows. First, one trainer is created for every dataset. Then, we can optimize the average loss by iterating through every task trainer in one epoch. The logic is shown in Code Snippet 3,

```
# Create a trainer for every task
>>> self.task_to_trainers = {}
>>> for task_name in tasks:
>>>     self.task_to_trainers[task_name] = Trainer(model, dataset, evaluator, ...)
# Optimize the average loss by iterating over all task trainers in each epoch.
>>> for epoch in range(1, 1 + epochs):
>>>     for task_name in task_list:
#           Each task trainer optimizes the loss of the task itself
>>>         task_trainer = self.task_to_trainers[task_name]
>>>         task_trainer.train_epoch()
```

Code Snippet 3: Implementation of multitask learning

To make this easy to use, we have wrapped the logic into a multitask trainer. As shown in Code Snippet 4, the user can create a multitask trainer by specifying a model and providing a list of datasets. After that, the user can train a multitask model using a single function.

```
>>> from ml_for_road_safety import MultitaskTrainer
# Specify the tasks that are combined in multitask learning
>>> task_list = ["MA_accident_classification", "MD_accident_classification", ...]
>>> task_datasets = {}; task_evaluators = {}
>>> for task_name in task_list:
>>>     state_name, data_type, task_type = task_name.split("_")
>>>     task_datasets[task_name] = TrafficAccidentDataset(state_name = state_name)
>>>     task_evaluators[task_name] = Evaluator(type=task_type)
# Initialize a trainer with a GNN model, multiple datasets, and multiple evaluators
>>> trainer = MultitaskTrainer(model, tasks = task_list,
    task_to_datasets=task_datasets, task_to_evaluators=task_evaluators, ...)
# Conduct multitask learning and evaluation for every task
>>> trainer.train()
```

Code Snippet 4: ML4RoadSafety Multitask Trainer

We use transfer learning to transfer knowledge from traffic volume information to traffic accident prediction. We implement this using our package by training traffic volume and traffic accident prediction simultaneously in a multitask model. As shown in Code Snippet 5, the user can create a multitask trainer to train a model on the accident and volume prediction tasks from one state.

```
>>> from ml_for_road_safety import MultitaskTrainer
# Specify the accident and volume prediction tasks from one state
>>> task_list = ["MA_accident_classification", "MA_volume_regression"]
>>> task_datasets = {}; task_evaluators = {}
>>> for task_name in task_list:
>>>     state_name, data_type, task_type = task_name.split("_")
>>>     task_datasets[task_name] = TrafficAccidentDataset(state_name = state_name)
>>>     task_evaluators[task_name] = Evaluator(type=task_type)
# Initialize a trainer with a GNN model and two tasks of accident and volume
    prediction.
>>> trainer = MultitaskTrainer(model, tasks = task_list,
    task_to_datasets=task_datasets, task_to_evaluators=task_evaluators, ...)
# Conduct multitask learning and evaluation for both tasks
>>> trainer.train()
```

Code Snippet 5: Implementation of transfer learning

### B.3.2 Applying graph contrastive learning on our dataset

Our package can be easily extended to incorporate graph contrastive learning methods in traffic accident prediction on our datasets. For example, we can implement graph contrastive learning [67] on our datasets with a few lines of code. As shown in Code Snippet 6, one can define a trainer for

contrastive learning by modifying the training loss in the base trainer class. Then, the user can use the trainer to conduct contrastive learning on our dataset.

```
>>> from ml_for_road_safety import Trainer
# Define a trainer for contrastive learning inherited from the base Trainer class
>>> class GraphContrastiveLearningTrainer(Trainer):
# Modify the training loss in the training logic
>>>     def train_epoch(self):
#           Define the contrastive loss
>>>         ...
>>>         loss = info_nce(outputs_1, outputs_2)
>>>         ...
# Initialize a contrastive learning trainer
>>> trainer = GraphContrastiveLearningTrainer(model, dataset, evaluator, ...)
# Conduct training and evaluation inside the trainer
>>> log = trainer.train()
```

Code Snippet 6: Implementation of graph contrastive learning

### B.3.3 Implementation of spatiotemporal graph neural networks

Next, our package supports the evaluation of spatiotemporal graph neural networks. For example, as shown in Code Snippet 7, the user can create a spatiotemporal model using our package, such as STGCN, by specifying the corresponding model name. Our package includes the implementation of spatiotemporal models from an open-sourced repository, `pytorch-geometric-temporal`. Then, the user can create a trainer object that directly trains the model on a given dataset.

```
>>> from ml_for_road_safety import Trainer, GNN
# Create a spatiotemporal model, e.g., STGCN, from an online implementation
>>> model = GNN(encoder = "stgcn", ...)
# Initialize a trainer with the model and specify use_time_series as True
>>> trainer = Trainer(model, dataset, evaluator, use_time_series=True)
# Conduct training and evaluation inside the trainer
>>> log = trainer.train()
```

Code Snippet 7: Training a spatiotemporal model

### B.3.4 Incorporating advanced multitask and transfer learning techniques

Lastly, our package can be easily extended to incorporate advanced multitask and transfer learning techniques. We describe two examples in the following.

For multitask learning, we consider two task grouping methods, which identify tasks that would benefit from training together and train them in one neural network as a group. These methods include task affinity grouping (TAG) [13] and approximating higher-order task groupings (HOA) [53]. Our package can be extended to incorporate these methods with a few lines of code. As shown in Code Snippet 8, the user can obtain the task grouping by using the task grouping methods. Then, the user can use the multitask trainer to train a multitask model on each group of tasks.

```
>>> from ml_for_road_safety import MultitaskTrainer
# Generate task groupings from previous task grouping methods, such as TAG or HOA
>>> task_list = ["DE_accident_classification", "IL_accident_classification",
    "MA_accident_classification", "MD_accident_classification", ...]
>>> task_groups = group_tasks(method = "hoa", task_list)
# Generated task grouping is a list of grouped tasks
>>> task_groups = [
        ["DE_accident_classification", "IL_accident_classification", ...],
        ["MA_accident_classification", "MD_accident_classification", ...],
        ["IA_accident_classification", "NV_accident_classification", ...]
    ]
>>> for group_task_list in task_groups:
>>>     task_datasets = {}; task_evaluators = {}
```

```
>>>    for task_name in group_task_list:
>>>        state_name, data_type, task_type = task_name.split("_")
>>>        task_datasets[task_name] = TrafficAccidentDataset(state_name)
>>>        task_evaluators[task_name] = Evaluator(task_type)
#      Initialize a trainer with the combined datasets of a group
>>>    trainer = MultitaskTrainer(model, tasks = group_task_list,
       task_to_datasets=task_datasets, task_to_evaluators=task_evaluators, ...)
#      Conduct multitask learning on one group of tasks
>>>    trainer.train()
```

Code Snippet 8: Training multitask learning models on groups of tasks

For transfer learning, we consider two regularization methods for fine-tuning a model trained on a source task to a target task. These methods include soft penalty (SP) and sharpness-aware minimization (SAM). Soft penalty regularizes the fine-tuned model distances to the initial model weights. Sharpness-aware minimization simultaneously minimizes loss value and loss sharpness with regard to the model weights. For example, as shown in Code Snippet 9, one can define a trainer to add the soft penalty loss by modifying the training loss in the base trainer class. Then, the user can use the trainer to fine-tune a model with a soft penalty on the fine-tuned distances.

```
>>> from ml_for_road_safety import Trainer
# Define a trainer for soft penalty inherited from the base Trainer class
>>> class SoftPenaltyTrainer(Trainer):
# Modify the training loss in the training logic
>>>    def train_epoch(self):
#          Combine the soft penalty loss with the cross-entropy loss
>>>        ...
>>>        loss = cross_entropy_loss + \
                lambda*add_soft_penalty(model, initial_state_dict)
>>>        ...
# Initialize a soft penalty trainer
>>> trainer = SoftPenaltyTrainer(model, dataset, evaluator, initial_state_dict, ...)
# Conduct training and evaluation inside the trainer
>>> log = trainer.train()
```

Code Snippet 9: Implementation of fine-tuning with soft penalties

We find that using these methods improves the test performance over simple methods by 0.6% on average over four states. More comprehensive evaluations of related methods are left for future work.

## C  Additional Related Works and Discussions

**Spatiotemporal graph neural networks.**  Previous works have proposed spatiotemporal graph neural networks to capture spatial and temporal dependencies for time series analysis on graph-structured data. DCRNN [39] captures the spatial dependency using bidirectional random walks on the graph and the temporal dependency using the encoder-decoder architecture with scheduled sampling. Instead of applying regular convolutional and recurrent units, STGCN [68] builds the model with complete convolutional structures, which enable much faster training speed with fewer parameters. Bai et al. [1] propose AGCRN with two adaptive modules for enhancing GNNs, including one module to capture node-specific patterns and another to infer the inter-dependencies among different traffic series automatically. Wu et al. [62] develop Graph WaveNet with an adaptive dependency matrix to capture the hidden spatial dependency in the data and a dilated 1D convolution component to handle very long sequences. We refer interested readers to a comprehensive survey [58] on spatio-temporal models.

**Graph contrastive learning.**  The use of contrastive learning on graph-structured data has been extensively explored in recent work [67]. A refined optimization algorithm is introduced in You et al. [66]. For spatiotemporal graph learning, Qu et al. [50] formulate a contrastive self-supervision method to predict fine-grained urban flows considering all spatial and temporal contrastive patterns.

Zhang et al. [71] build a heterogeneous graph neural architecture to capture the multi-view region dependencies concerning POI semantics, mobility flow patterns, and geographical distances.

**Explainability in graph neural networks.** Yuan et al. [69] provide unified and taxonomic explanations regarding the importance of a node/an edge/a subgraph in a graph neural network, etc. In our leave-one-out analysis, because we are interested in which categories of information (graph structural vs. weather vs. traffic volume) are most useful, our analysis can be viewed as a first-order explanation of the importance of graph structural features. Further applying the methods of Yuan et al. [69] to explain our findings is a research question for future studies.

**Multitask and transfer learning.** Our methods for conducting multitask learning are based on recent theoretical developments regarding negative transfer in multitask learning, thus calling for more robust procedures [64]. In particular, Li et al. [36] propose surrogate modeling to approximate multitask predictions. Li et al. [35] introduce a boosting procedure as an ensemble method for multitask learning on graph-structured data. Our methods for transfer learning are based on recent developments for robust fine-tuning [37, 33]. Ju et al. [32] develop a spectrally-normalized generalization bound for graph neural networks and design a noise-stability optimization algorithm for improved fine-tuning.

**Development of our dataset.** Our dataset includes the road network features and weather information for all the states in the US. The bottleneck is the traffic volume and the accident records. For the eight states in our current dataset, both types of data are published by the Department of Transportation on the respective state's website. See the links to each state's government website in Table 6.

For the other states, we have checked their Department of Transportation websites, and we could not find detailed data, including accidents and traffic volume (like the eight states we currently have). Once the data is updated, we would be happy to update our dataset as well.

For a few states, for example, California and New York, the traffic volume data and accident information are both available for a few counties through their transportation departments, such as Los Angeles and New York City. For New York City, we have collected 2.02 million accident records from 2012 to 2023, including the latitude and longitude of each accident. For California, we have 0.4 million Motor Vehicle Crashes from 2016 to 2021. However, these do not have the latitude and longitude information, so we cannot match a record to a particular edge/node of the network.

