# OpenReview forum: "Graph Neural Networks for Road Safety Modeling: Datasets and Evaluations for Accident Analysis"
_NeurIPS.cc/2023/Track/Datasets_and_Benchmarks — NeurIPS 2023 Datasets and Benchmarks Poster_

### Official Review · Reviewer_F2Hc · 2023-07-10
**A good improvement on existing datasets.**

**Rating:** 6
**Confidence:** 5
**Clarity:** The overall readability of the paper …

**Strengths:**

- well organize and refine the current dataset in a unified format for easy use.
- a comprehensive evaluation of popular graph neural networks.
- constructive discussion for the future work

**Additional Feedback:**

N/A

**Correctness:**

The details of the used multitask learning are not clear, and so the transfer learning.

**Documentation:**

The code is provides with little documentation, even it might not be hard to understand the use of it.

**Limitations:**

The attacker might employ this dataset to improve the skill of attacking the graph neural network as the prediction model. So it might make graph neural networks predict less or no accident number, and release the preparation for the emergence scenario.

**Opportunities For Improvement:**

- It's better to include a discussion or evaluation of spatio-temporal graph neural networks.
- In 153, `annual level` regression seems not practical in real-world scenarios. The author may want to domain the use of annual level regression.
- the `weather` attribute looks promising, however, it is highly uncertain when making a prediction, but it is given (no uncertainty) when doing training.

**Relation To Prior Work:**

One concern is the relation to `spatio-temporal graph neural network` is not clear, as they should be close.

**Summary And Contributions:**

- create a 9-million-record traffic accident dataset from eight states' official reports, with several additional features beyond the existing dataset.
- perform a comprehensive evaluation with graph neural networks
- describe the findings and create a package for simply reusing the new datasets and library.

---

> ### Author Response · Authors · 2023-08-22
> **Response to Reviewer F2Hc 1/2**
>
> Thanks very much for the reviewer’s constructive comments about our work. We respond to the reviewer’s comments below.
>
> **>>> “It's better to include a discussion or evaluation of spatio-temporal graph neural networks.”**
>
> Thanks for the suggestion. We added a more detailed discussion of spatiotemporal GNN now, including a few more recent models such as STGCN, Graph WaveNet, and AGCRN. We have also done evaluations on our datasets, and here is the result:
>
> | Model (# parameters) | DE           | MA           | MD           | NV           |
> |--------------------------|--------------|--------------|--------------|--------------|
> | Labeling Rate  | 0.23  |  0.10  | 0.15  | 0.12  |
> | Training records         | 112,670      | 1,265,895    | 341,902      | 101,975      |
> | | | | | |
> | GraphSAGE (115K)    | 87.6±0.1     | 81.8±0.1     | 87.5±0.0     | 91.6±0.9     |
> | AGCRN (219K)           | 86.0±0.2     | 82.1±0.2     | 88.5±0.1     | 90.7±0.2     |
> | STGCN (638K)           | 85.4±0.1     | 81.9±0.3     | 88.7±0.1     | 91.5±0.3     |
> | Graph WaveNet (1079K)  | 85.0±0.2     | 81.9±0.5     | 87.9±0.1     | 90.6±0.2     |
> | DCRNN (1388K)         | 81.2±1.2     | 70.5±0.1     | 84.5±0.3     | 90.5±0.7     |
>
> Our finding is the performance of GraphSAGE is still comparable to there three GNNs. Our explanation for this is our dataset is quite sparse in terms of the accident labels. These results and the discussions are added in Appendix D.1 and E.3 of the updated supplementary materials.
>
> **>>> “In 153, annual level regression seems not practical in real-world scenarios. The author may want to domain the use of annual level regression.”**
>
> First of all, this annual level regression is applied to the traffic volume (AADT) labels, specifically, given a graph, we want to predict the annual average daily traffic (AADT) on a particular road. The reason why this regression task is at an annual level is because the AADT information, which we extract from the Department of Transportation at each state, is at an annual level. Therefore, it is not possible to set up this task at other granularities. Notice however that our traffic accident prediction model is set up at a monthly level.
>
> Since the AADT labels are sparse, i.e., they cover less than 4% of all roads (see Table 1), the annual level regression task can be used to estimate the unobserved AADT values of the roads which are currently not reported.
>
> **>>> “The weather attribute looks promising, however, it is highly uncertain when making a prediction, but it is given (no uncertainty) when doing training.”**
>
> For each coordinate (latitude, longitude) in the traffic network, the historical weather data is collected from the nearest meteorological station. Note that this is historical data, so the uncertainty would be low, because it is already the reported attributes. We are not predicting these values.
>
> **>>> “The attacker might employ this dataset to improve the skill of attacking the graph neural network as the prediction model. So it might make graph neural networks predict less or no accident number, and release the preparation for the emergence scenario.”**
>
> Thanks for a very interesting perspective. We plan to release our dataset and the evaluation codes after the review process and hope future work could further explore these topics, such as protecting the network from adversarial attacks. At the same time, all of the data we have gathered are from the public domain, but we hope our unified framework could further guide the development in this area.

---

> ### Author Response · Authors · 2023-08-22
> **Response to Reviewer F2Hc 2/2**
>
> **>>> “The details of the used multitask learning are not clear, and so the transfer learning.”**
>
> Thanks for the suggestion; We have added a section to describe our practical implementation of multitask and transfer learning. For multitask learning, we train a single GNN with the combined data from multiple states. To implement this, first, we create one trainer for every state dataset. Then, we optimize the average loss by iterating through every trainer in each epoch; here are the snippets:
>
> ``` Python
> # Create a trainer for every task
> >>> self.task_to_trainers = {}
> >>> for task_name in tasks:
> >>>     self.task_to_trainers[task_name] = Trainer(model, dataset, evaluator, ...)
> # Optimize the average loss by iterating over all task trainers in each epoch.
> >>> for epoch in range(1, 1 + epochs):
> >>>     for task_name in task_list:
> #           Each task trainer optimizes the loss of the task itself
> >>>         task_trainer = self.task_to_trainers[task_name]
> >>>         task_trainer.train_epoch()
> ```
>
> To make this easy to use, we have wrapped the logic into a multitask trainer. One can create a multitask trainer by specifying a model and providing a list of datasets. Then, our code will train a multitask model in a single function.
>
> ``` Python
> >>> from ml_for_road_safety import MultitaskTrainer
> # Specify the tasks that are combined in multitask learning
> >>> task_list = ["MA_accident_classification", "MD_accident_classification", ...]
> >>> task_datasets = {}; task_evaluators = {}
> >>> for task_name in task_list:
> >>>     state_name, data_type, task_type = task_name.split("_")
> >>>     task_datasets[task_name] =  TrafficAccidentDataset(state_name = state_name)
> >>>     task_evaluators[task_name] = Evaluator(type=task_type)
> # Initialize a trainer with a GNN model, multiple datasets, and multiple evaluators
> >>> trainer = MultitaskTrainer(model, tasks = task_list, task_to_datasets=task_datasets, task_to_evaluators=task_evaluators, ...)
> # Conduct multitask learning and evaluation for every task
> >>> trainer.train()
> ```
>
> For transfer learning, we train one GNN for predicting the traffic volume labels and the accident labels simultaneously. This is implemented in a multitask trainer for one state as follows:
>
> ``` Python
> >>> from ml_for_road_safety import MultitaskTrainer
> # Specify the accident and volume prediction tasks from one state
> >>> task_list = ["MA_accident_classification", "MA_volume_regression"]
> >>> task_datasets = {}; task_evaluators = {}
> >>> for task_name in task_list:
> >>>     state_name, data_type, task_type = task_name.split("_")
> >>>     task_datasets[task_name] =  TrafficAccidentDataset(state_name = state_name)
> >>>     task_evaluators[task_name] = Evaluator(type=task_type)
> # Initialize a trainer with a GNN model and two tasks of accident and volume prediction.
> >>> trainer = MultitaskTrainer(model, tasks = task_list, task_to_datasets=task_datasets, task_to_evaluators=task_evaluators, ...)
> # Conduct multitask learning and evaluation for both tasks
> >>> trainer.train()
> ```

---

### Official Review · Reviewer_rFa1 · 2023-07-20
**Review of Submission 422**

**Rating:** 6
**Confidence:** 4

**Strengths:**

- The unified large-scale accident dataset enables researchers to conduct more comprehensive evaluations and delve into further in-depth studies.
- The experiment design shows diversity, including the evaluation of multiple model architectures, and multiple training paradigms.
- The code and dataset package greatly facilitates the development of novel methods for accident prediction.

**Additional Feedback:**

NA

**Clarity:**

The writing of this paper is fairly clear. Some clarity concerns are listed as below.

- In line 148, why combining traffic flow information as an auxiliary task is a transfer learning technique?
- There appears to be an inconsistency in the paper regarding the joint learning of accident and volume prediction. While in line 187, the authors mention their intention to jointly learn these two tasks, in line 235, the effectiveness of traffic volume is evaluated in a fine-tuning manner.
- In line 214, the use of 'improves generalization' is misleading, as there is only evidence of volume information enhancing the MAE or AUROC of accident prediction.

**Correctness:**

The paper achieves a fairly good correctness in dataset construction, methodology and experiment design & analysis.


**Documentation:**

The paper provides a package with detailed elaboration on how to use the dataset to facilitate the development of new methods.


**Limitations:**

See Opportunities For Improvement

**Opportunities For Improvement:**

- The motivation of collecting such a dataset is not clear.
  - In lines 34 to 45, the paper argues that the construction of this dataset aims to facilitate a comprehensive evaluation of existing deep learning methods. However, the reasons why evaluating on datasets collected in previous works fail to showcase the potential of existing deep learning methods remain unclear.
  - While the authors emphasize the large-scale nature of the dataset, the extent of its scalability is insufficiently explained. Merely increasing the number of traffic records from 2 million (as mentioned in reference [37] in the paper) to 9 million is not persuasive enough. Additionally, the dataset's spatial range lacks comparison with other existing datasets, making it difficult to assess its significance in terms of scale.
- The novelty of the dataset construction approach presented in the paper raises skepticism. The paper lacks a thorough discussion of existing spatio-temporal benchmarks and fails to provide a comprehensive comparison between the construction methods employed in other works and the methodology used for this dataset.
- The motivation of the experiment design is insufficiently discussed. It is confusing that why further test multi-task learning and transfer training beyond several GNN baselines. There are tons of ways to perform multi-task learning or transfer learning, why do the authors only adopt the two simple ways mentioned in the paper?
- The experiments fall short in terms of including SOTA STGNNs such as GraphWavenet, AGCRN, etc. Furthermore, it fails to incorporate existing SOTA works on traffic accident prediction.
- The experimental findings are superficial and insufficient.
  - In section 3.2, takeaway (1), the authors only present the effectiveness of GNNs without an analysis of why GNNs works well. For example, the authors do not explain the reasons behind the superior performance of GraphSAGE compared to other GNNs and even DCRNN, which may appear counterintuitive and requires further elaboration.
  - In section 3.2, takeaway (2) and (3), the authors' findings are natural and common. The multitask learning and incorporate volume information for joint training have already been proved to be effective in various spatio-temporal applications.
  - In section 3.3, the leave-one-out analysis of different features is outdated considering there are so many methods to generate post-hoc explanations for GNNs [1].

[1] Yuan et al. Explainability in Graph Neural Networks: A Taxonomic Survey. In TPAMI 2022.

**Relation To Prior Work:**

- The constructed dataset exhibits an increase in the number of data records as well as an improved diversity in spatial distribution. However, the paper fails to provide a detailed comparison with datasets used in previous accident prediction works, making it difficult to assess the significance of these improvements.
- The paper solely compares the constructed accident dataset with other graph benchmarks and overlooks the comparison with existing spatio-temporal datasets.

**Summary And Contributions:**

The paper constructs a large-scale traffic accident dataset across 8 US states, totaling over 9 million records spanning up to 20 years, which is currently the largest traffic accident dataset. For each state, road network graphs are generated using OpenStreetMap data. Node features include weather, location, connectivity. Edge features include road length, type. In addition, Annual Average Daily Traffic volume (AADT) data is aligned to road edges where available.

The paper comprehensively evaluates predictive performance of graph learning methods including network embedding methods, graph neural networks and spatio-temporal graph neural networks and find GNNs can achieve better performance. The authors also explore the potential of leveraging cross-state trend similarities by conducting Multi-Task Learning (MTL) on data from all states with more labels, and observe that MTL outperforms single task models. Furthermore, Traffic volume prediction is incorporated as an auxiliary task to explore whether it can improve accident prediction, and the results show positive influences.

The paper concludes by discussing the broader implications of the constructed dataset and the extensive experiments. The authors also provide an easy-to-use package that enables researchers to explore and build upon the dataset.

---

> ### Author Response · Authors · 2023-08-22
> **Response to Reviewer rFa1 1/5**
>
> We are grateful for the reviewer’s detailed comments, including carefully reading our paper and giving constructive suggestions. We have taken the reviewer’s suggestions for improvement in the rebuttal and here is our response.
>
> **>>> “The motivation of collecting such a dataset is not clear. Additionally, the dataset's spatial range lacks comparison with other existing datasets.”**
>
> Our motivation for constructing this dataset is about how we can apply machine learning to aid road safety, which is an important real-world problem; See, e.g., CDC’s reports [1,2].
>
> There are lots of developments in spatiotemporal graph learning, and we believe our work from the perspective of traffic accident analysis could be a nice complement to existing works. Existing datasets about traffic accident prediction [37] use data collected from streaming APIs that provide accident information for certain times of the day (e.g., during rush hours), and there is some discussion about reporting errors in the start and end time of this data. To ensure the validity of our analysis, we think it is important to ensure the quality of the accident data; Ours is collected directly from the official reports of the Department of Transportation–we think this is the key merit of our dataset.
>
> Below is a detailed comparison between our dataset and two existing datasets that also consider traffic accident prediction, including the spatial range of each dataset.
>
> |             | Huang et al., (2023) | Yuan et al., (2018) | Our Dataset |
> |-------------|---------------------|-----------------------|-------------|
> | Data Source | Based on a work by Moosavi et al. (2019), sourced from Microsoft Bing Map Traffic | Department of Transportation’s official accident reports | Department of Transportation’s official accident reports |
> | Coverage    | 2.8 million accident records, across 13 states, for 2016-2021 | only for Iowa, for 2006-2013 | 9 million records, across 8 states, for a maximum of 20 years up to 2023 |
> | Prediction Tasks | Node-level classification, evaluated on AUROC | Spatial grid-level regression, evaluated under MAE | Edge-level classification/regression, evaluated under both MAE/AUROC |
> | Features    | Road network features | Averaged grid-level traffic volume, rainfall features alongside the grid-level features | Traffic volume and weather features alongside the road network features |
>
> We can see that our dataset has more coverage than the two datasets under comparison. We apologize if this does not come across as clearly in the initial submission, and we will revise the paper to make the comparison more clear.
>
> [1] [https://www.cdc.gov/transportationsafety/pdf/CDC-DIP_At-a-Glance_Transportation_508.pdf](https://www.cdc.gov/transportationsafety/pdf/CDC-DIP_At-a-Glance_Transportation_508.pdf)
>
> [2] [https://www.cdc.gov/transportationsafety/index.html](https://www.cdc.gov/transportationsafety/index.html)
>
> Yuan, Z., Zhou, X., & Yang, T. (2018, July). Hetero-convlstm: A deep learning approach to traffic accident prediction on heterogeneous spatio-temporal data. KDD.
>
> Huang, B., Hooi, B., & Shu, K. (2023). TAP: A Comprehensive Data Repository for Traffic Accident Prediction in Road Networks. arXiv preprint arXiv:2304.08640.

---

> ### Author Response · Authors · 2023-08-22
> **Response to Reviewer rFa1 2/5**
>
> **>>> “The novelty of the dataset construction approach presented in the paper raises skepticism. The paper lacks a thorough discussion of existing spatio-temporal benchmarks and fails to provide a comprehensive comparison between the construction methods employed in other works and the methodology used for this dataset.”**
>
> Thanks for the detailed feedback. Since our focus is about traffic accident prediction, we think this could be a nice complement to existing spatiotemporal benchmarks. Just to give a few concrete examples (which are by no means exhaustive), here is a table to compare our dataset with several spatiotemporal datasets:
>
> |          | METR-LA (Li et al., 2017) | PEMS-Bay (Li et al., 2017) | Taxi NYC (Sun et al., 2020) | Didi Chuxing (Li et al., 2018) | Our Dataset |
> |----------|-------------------------------|-------------------------------|---------------------------------|-------------------------------|-------------|
> | Prediction Targets  | Traffic speed and volume      | Traffic speed and volume      | # Rides in a region            | Trip time                     | Traffic accident |
> | Coverage | 6 million for 4 months from Mar. 2012 to Jun. 2012 | 16 million for 6 months from Jan. 2017 to May 2017 | 35 million over 5 years from Jan. 2011 to Jun. 2016 | 61.4 million over six months from May. 2017 to Oct. 2017 | 9 million in 8 US states, for up to 20 years from 2002 to 2023. |
> | Data Source | Highway loop detectors | CalTrans Performance Measurement System | Taxi GPS data for NYC | Didi Chuxing Beijing | Department of Transportation |
>
> In terms of construction methods, both METR-LA and PEMS-BAY are collected from loop detectors on the highways of California Transportation Agencies. NYC Open Data and DiDi Chuxing collect taxi traffic information from their platforms.
>
> For our dataset, we went through each state’s Department of Transportation website and found the corresponding data sheets when they are available (some are in multiple PDFs), and then we extracted them into a standardized format. We construct the road network features from OpenStreetMap. We collected traffic volume (AADT) information from the Department of Transportation of each state as well.
>
> We again apologize if these do not come across as clearly in the initial submission, and we will revise the paper to highlight the comparison more clearly.

---

> ### Author Response · Authors · 2023-08-22
> **Response to Reviewer rFa1 3/5**
>
> **>>> “The motivation of the experiment design is insufficiently discussed. It is confusing that why further test multi-task learning and transfer training beyond several GNN baselines. There are tons of ways to perform multi-task learning or transfer learning, why do the authors only adopt the two simple ways mentioned in the paper?”**
>
> Our motivation for multitask learning is that combining the data from multiple states can help the prediction states with fewer labels. The labeling rate of our datasets, which is the percentage of edges with a positive label/accident occurrence, is quite low; Thus, it is conceivable that multitask learning can be helpful. The motivation for transfer learning is similar, since the traffic volume labels provide extra information to help with accident prediction.
>
> Just to be concrete, here is the labeling rate of each state (from Table 2):
>
> | States         | DE    | IA    | IL    | MA    | MD    | MN    | MT    | NV    |
> |----------------|-------|-------|-------|-------|-------|-------|-------|-------|
> | Labeling Rate  | 0.23  | 0.07  | 0.14  | 0.10  | 0.15  | 0.05  | 0.05  | 0.12  |
>
> Following the reviewer’s suggestion, we conducted additional experiments using advanced multitask and transfer learning methods. We emphasize that because we want to focus on the dataset and provide a easy-to-use package that allows easy extensions to more method evaluations, we choose to focus on the two simple ways. It is a research question to comprehensively evaluate more advanced multitask and transfer learning methods on our dataset for future work.
>
> Just to give a few examples, we extend our package to implement two recent multitask learning methods, including task affinity grouping (TAG) by Fifty et al. (2021) and another grouping method (HOA) by Standley et al. (2021). We can directly train multitask learning models on task groups based on implementations online as follows:
>
> ``` Python
> >>> from ml_for_road_safety import MultitaskTrainer
> # Generate task groupings with task grouping methods, such as HOA
> >>> task_list = ["DE_accident_classification", "IL_accident_classification", ...]
> >>> task_groups = group_tasks(method = "hoa", task_list)
> # Generated task grouping is a list of grouped tasks
> >>> task_groups = [
>         ["DE_accident_classification", "IL_accident_classification", ...],
>         ["MA_accident_classification", "MD_accident_classification", ...],
>         ["IA_accident_classification", "NV_accident_classification", ...], ...
>     ]
> >>> for group_task_list in task_groups:
> >>>     task_datasets = {}; task_evaluators = {}
> >>>     for task_name in group_task_list:
> >>>         state_name, data_type, task_type = task_name.split("_")
> >>>         task_datasets[task_name] =  TrafficAccidentDataset(state_name)
> >>>         task_evaluators[task_name] = Evaluator(task_type)
> #       Initialize a trainer with the combined datasets of a group
> >>>     trainer = MultitaskTrainer(model, tasks = group_task_list,
>         task_to_datasets=task_datasets, task_to_evaluators=task_evaluators, ...)
> #       Conduct multitask learning on one group of tasks
> >>>     trainer.train()
> ```
> Here are some examples of using more advanced fine-tuning and transfer learning methods, including $\ell_2$-regularized soft penalty (SP) by Li et al. (2018) and Sharpness-Aware Minimization (SAM) by Foret et al. (2021). For example, we implement fine-tuning with soft penalty by modifying the training loss in the predefined training logic of our package.
>
> ``` Python
> >>> from ml_for_road_safety import Trainer
> # Define a trainer for soft penalty inherited from the base Trainer class
> >>> class SoftPenaltyTrainer(Trainer):
> # Modify the training loss in the training logic
> >>>     def train_epoch(self):
> #           Combine the soft penalty loss with the cross-entropy loss
> >>>         ...
> >>>         loss = cross_entropy_loss + \
>                    lambda*add_soft_penalty(model, initial_state_dict)
> >>>         ...
> # Initialize a soft penalty trainer
> >>> trainer = SoftPenaltyTrainer(model, dataset, evaluator, initial_state_dict, ...)
> # Conduct training and evaluation inside the trainer
> >>> log = trainer.train()
> ```

---

> ### Author Response · Authors · 2023-08-22
> **Response to Reviewer rFa1 4/5**
>
> We report our preliminary findings below: Using more advanced methods improves upon the two simple ways we have reported by 0.6% averaged over four states. These results have been added to Appendix E.5 of the updated supplementary materials. We leave a more comprehensive evaluation of multitask and transfer learning methods to future work.
>
> |      | DE            | MA            | MD            | NV            |
> |---------------|---------------|---------------|---------------|---------------|
> | STL           | 87.6±0.1      | 81.8±0.1      | 87.5±0.0      | 91.6±0.9      |
> | | | | | |
> | MTL           | 87.7±0.1      | 79.6±0.1      | 88.1±0.1      | 90.3±0.2      |
> | MTL-FT        | 87.8±0.3      | 81.9±0.3      | 88.1±0.1      | 91.8±0.2      |
> | MTL-HOA       | 88.5±0.2      | 82.9±0.5      | 88.6±0.8      | 93.0±0.6      |
> | MTL-TAG       | 88.3±0.2      | 82.6±0.4      | 88.4±0.3      | 92.5±0.3      |
> | | | | | |
> | TL            | 87.3±0.2      | 82.6±0.2      | 87.9±0.4      | 92.8±0.1      |
> | TL-SP | 88.2±0.2    | 81.3±0.3      | 88.4±0.4      | 92.9±0.2      |
> | TL-SAM        | 88.3±0.2      | 82.1±0.2      | 88.2±0.6      | 93.0±0.5      |
>
> Fifty, Chris, et al. "Efficiently identifying task groupings for multi-task learning." Advances in Neural Information Processing Systems 34 (2021): 27503-27516.
>
> Standley, Trevor, et al. "Which tasks should be learned together in multi-task learning?." International Conference on Machine Learning. PMLR, 2020.
>
> Li, Xuhong, et al. "Explicit inductive bias for transfer learning with convolutional networks." International Conference on Machine Learning. PMLR, 2018.
>
> Foret, Pierre, et al. "Sharpness-aware minimization for efficiently improving generalization." ICLR (2021).

---

> ### Author Response · Authors · 2023-08-22
> **Response to Reviewer rFa1 5/5**
>
> **>>> “The experiments fall short in terms of including SOTA STGNNs such as GraphWavenet, AGCRN, etc. Furthermore, it fails to incorporate existing SOTA works on traffic accident prediction. In section 3.2, takeaway (1), the authors only present the effectiveness of GNNs without an analysis of why GNNs works well. For example, the authors do not explain the reasons behind the superior performance of GraphSAGE compared to other GNNs and even DCRNN.”**
>
> Thanks for suggesting these baselines. We conducted additional evaluations using the models suggested by the reviewer and report the results below:
>
> | Model (# parameters) | DE           | MA           | MD           | NV           |
> |--------------------------|--------------|--------------|--------------|--------------|
> | Labeling Rate  | 0.23  |  0.10  | 0.15  | 0.12  |
> | Training records         | 112,670      | 1,265,895    | 341,902      | 101,975      |
> | | | | | |
> | GraphSAGE (115K)    | 87.6±0.1     | 81.8±0.1     | 87.5±0.0     | 91.6±0.9     |
> | AGCRN (219K)           | 86.0±0.2     | 82.1±0.2     | 88.5±0.1     | 90.7±0.2     |
> | STGCN (638K)           | 85.4±0.1     | 81.9±0.3     | 88.7±0.1     | 91.5±0.3     |
> | Graph WaveNet (1079K)  | 85.0±0.2     | 81.9±0.5     | 87.9±0.1     | 90.6±0.2     |
> | DCRNN (1388K)         | 81.2±1.2     | 70.5±0.1     | 84.5±0.3     | 90.5±0.7     |
>
> We clarify that we are merely saying that GraphSAGE achieves comparable performance to the other models, and this is reinforced by the above comparison. The reason is that the labeling rate of our dataset is very low (around or below 0.2%). Spatiotemporal GNNs have more trainable parameters than GraphSAGE; thus, they need more training labels in order to fit the model well.
>
> The reason why GNNs work well is because there are transferable graph structures across states for accident occurrences. For example, intersections of roads, highways tend to have more accidents; This is highlighted by Figure 1a and 1c between MA and NV. Our finding in the context of traffic accident prediction could be a complement to existing works; this is justified by the cross-state transfers we showed in Figure 4.
>
> We thank the reviewer for these great comments and we will revise the paper to incorporate these discussions.
>
> **>>> “In section 3.3, the leave-one-out analysis of different features is outdated considering there are so many methods to generate post-hoc explanations for GNNs [1].”**
>
> Thanks for bringing this reference to our attention. We note that Yuan et al. (2022) (and the references therein such as GNNExplainer) can further provide unified and taxonomic explanations regarding the importance of a node/an edge/or a subgraph etc. In our leave-one-out analysis, we are interested in which category (graph structural vs. weather vs. traffic volume) are most helpful. Our analysis can be viewed as a first-order explanation to the importance of graph structural features. Future work could further look into node/edge level explanations. We have also added Yuan et al. (2022) to the related work in the updated materials.
>
> **>>> “In line 148, why combining traffic flow information as an auxiliary task is a transfer learning technique? While in line 187, the authors mention their intention to jointly learn these two tasks, in line 235, the effectiveness of traffic volume is evaluated in a fine-tuning manner. In line 214, the use of 'improves generalization' is misleading, as there is only evidence of volume information enhancing the MAE or AUROC of accident prediction.”**
>
> The traffic flow labels (AADT) provide auxiliary information about the traffic network that is not provided by the accident labels. Therefore, combining these traffic flow labels as an auxiliary task can be viewed as some sort of “transfer learning” from task A (traffic flow prediction) to task B (accident label prediction).
>
> In line 235, we want to conduct an ablation study to provide extra evidence that there is a positive transfer from traffic flow prediction to accident label prediction. This extra evidence is provided in the fine-tuning setting, so we are merely saying that we observe positive transfers for both settings of (1) joint learning followed by fine-tuning, (2) fine-tuning task A on task B. See also Zhu et al. (2021) for a related work about transfer learning of graph neural networks.
>
> Here improving generalization just means improving test performance. We have clarified this in the paper. Thanks for noticing this issue.
>
> Zhu, Qi, et al. "Transfer learning of graph neural networks with ego-graph information maximization." Advances in Neural Information Processing Systems 34 (2021): 1766-1779.

---

### Official Review · Reviewer_nmaH · 2023-07-23
**The paper co	The overall structure and logic of the article are quite comprehensive. The authors provide a complete workflow for model construction and experimentation, including details about baseline methods, graph neural network models, multitask learning, and transfer learning techniques, along with their specific implementations. 	The article offers detailed explanations of framework choices and provides links to the code. Such comprehensive descriptions are beneficial for readers to understand the experimental design and replicate the author's research.uld be accept**

**Rating:** 6
**Confidence:** 3
**Clarity:** Yes

**Strengths:**

	The overall structure and logic of the article are quite comprehensive. The authors provide a complete workflow for model construction and experimentation, including details about baseline methods, graph neural network models, multitask learning, and transfer learning techniques, along with their specific implementations.
	The article offers detailed explanations of framework choices and provides links to the code. Such comprehensive descriptions are beneficial for readers to understand the experimental design and replicate the author's research.


**Additional Feedback:**

N/A

**Correctness:**

Are the claims made in the submission correct?  Yes
If the submission is a dataset, is it constructed in a sound way?  Yes


**Documentation:**

The article presents a complete documentation and experimental process.

**Limitations:**

The article does not provide a clear and detailed explanation of its own limitations.

**Opportunities For Improvement:**

	In the methodology section, when introducing the concepts of multitask learning and transfer learning, further elaboration on how they are practically implemented in the actual model and the specific structure of these methods would be helpful. Currently, the implementation details and structures of the specific models remain unclear.
	In the experimental section, there is limited logical explanation regarding framework choices and model implementations. For example, the rationale behind selecting the feature encoder and the introduction of spatiotemporal models to address specific issues are not clearly explained.
	During the discussion of related works, the author provides relatively little information on the gaps and advantages of the current research compared to existing works. Including more comparative analysis would help readers understand the significance and value of the current research.
	After citing multiple references, the author does not evaluate or analyze these papers, merely listing them. To improve the understanding of the importance of these references and their relevance to the current research, it would be beneficial to include appropriate evaluations or analyses within the article.


**Relation To Prior Work:**

No

**Summary And Contributions:**

This paper explores the application of graph neural networks (GNNs) for traffic accident risk modeling, aiming to predict accident occurrences on road networks using graph-structural and related features. The importance of this modeling is recognized due to the economic costs and safety implications of motor vehicle crashes. The study presents a new dataset with over 9 million traffic accident records spanning eight states, and through evaluation, shows that GNNs, particularly GraphSAGE, can predict accident counts with a mean absolute error of 22% and whether an accident occurs with over 87% AUROC, across the states. The paper makes significant contributions by providing a unified dataset, demonstrating the effectiveness of GNNs, and discussing implications for policy interventions.

---

> ### Author Response · Authors · 2023-08-22
> **Response to Reviewer nmaH 1/2**
>
> Thanks for the constructive comments. We are glad that the reviewer appreciated the comprehensive structure of our work. We have taken the reviewer’s suggestion for improvement into our revision, and here is our response to each comment.
>
> **>> “In the methodology section, when introducing the concepts of multitask learning and transfer learning, further elaboration on how they are practically implemented in the actual model and the specific structure of these methods would be helpful.”**
>
> Thanks for the suggestion; We have added a section to describe our practical implementation of multitask and transfer learning. For multitask learning, we train a single GNN with the combined data from multiple states. To implement this, first, we create one trainer for every state dataset. Then, we optimize the average loss by iterating through every trainer in each epoch; here are the snippets from our package:
>
> ``` Python
> # Create a trainer for every task
> >>> self.task_to_trainers = {}
> >>> for task_name in tasks:
> >>>     self.task_to_trainers[task_name] = Trainer(model, dataset, evaluator, ...)
> # Optimize the average loss by iterating over all task trainers in each epoch.
> >>> for epoch in range(1, 1 + epochs):
> >>>     for task_name in task_list:
> #           Each task trainer optimizes the loss of the task itself
> >>>         task_trainer = self.task_to_trainers[task_name]
> >>>         task_trainer.train_epoch()
> ```
>
> To make this easy to use, we have wrapped the logic into a multitask trainer. One can create a multitask trainer by specifying a model and providing a list of datasets. Then, our code will train a multitask model in a single function.
>
> ``` Python
> >>> from ml_for_road_safety import MultitaskTrainer
> # Specify the tasks that are combined in multitask learning
> >>> task_list = ["MA_accident_classification", "MD_accident_classification", ...]
> >>> task_datasets = {}; task_evaluators = {}
> >>> for task_name in task_list:
> >>>     state_name, data_type, task_type = task_name.split("_")
> >>>     task_datasets[task_name] =  TrafficAccidentDataset(state_name = state_name)
> >>>     task_evaluators[task_name] = Evaluator(type=task_type)
> # Initialize a trainer with a GNN model, multiple datasets, and multiple evaluators
> >>> trainer = MultitaskTrainer(model, tasks = task_list, task_to_datasets=task_datasets, task_to_evaluators=task_evaluators, ...)
> # Conduct multitask learning and evaluation for every task
> >>> trainer.train()
> ```
>
> For transfer learning, we train one GNN to predict the traffic volume labels and the accident labels simultaneously. This is implemented in a multitask trainer for one state as follows:
>
> ``` Python
> >>> from ml_for_road_safety import MultitaskTrainer
> # Specify the accident and volume prediction tasks from one state
> >>> task_list = ["MA_accident_classification", "MA_volume_regression"]
> >>> task_datasets = {}; task_evaluators = {}
> >>> for task_name in task_list:
> >>>     state_name, data_type, task_type = task_name.split("_")
> >>>     task_datasets[task_name] =  TrafficAccidentDataset(state_name = state_name)
> >>>     task_evaluators[task_name] = Evaluator(type=task_type)
> # Initialize a trainer with a GNN model and two tasks of accident and volume prediction.
> >>> trainer = MultitaskTrainer(model, tasks = task_list, task_to_datasets=task_datasets, task_to_evaluators=task_evaluators, ...)
> # Conduct multitask learning and evaluation for both tasks
> >>> trainer.train()
> ```
>
> **>> “In the experimental section, there is limited logical explanation regarding framework choices and model implementations. For example, the rationale behind selecting the feature encoder and the introduction of spatiotemporal models to address specific issues are not clearly explained.”**
>
> Thanks for the feedback. There are four types of baselines in our experiments, including
>
> - A two-layer perceptron model trained directly with the node features: This tests using node features without network structures.
>
> - Node embeddings, including Node2Vec and DeepWalk. This assesses the benefits of using these node features inside the GNN.
>
> - Different GNN architectures, including GCN, GraphSAGE, and GIN.
>
> - Spatiotemporal GNNs such as DCRNN. These models generally have more trainable parameters and are more complex than the ones above, but they are designed to leverage time serial correlations in the underlying graphs.
>
> We hope these clarify the logic, and we will revise the text to make this more clear in the revised paper.

---

> > ### Comment · Reviewer_nmaH · 2023-08-23
> >
> > Thanks for the response. I think the paper is Ok to be accepted.

---

> > > ### Author Response · Authors · 2023-08-28
> > > **Thank you**
> > >
> > > Thanks very much for your prompt response. Your suggestions for improvement on our work have been very helpful. We greatly appreciate your positive opinion of our work.

---

> ### Author Response · Authors · 2023-08-22
> **Response to Reviewer nmaH 2/2**
>
> **>> “During the discussion of related works, the author provides relatively little information on the gaps and advantages of the current research compared to existing works. Including more comparative analysis would help readers understand the significance and value of the current research.”**
>
> We are grateful for the reviewer’s detailed feedback. Since our main contribution is the development of a new traffic accident dataset, we give more detailed comparison between our dataset and two existing datasets below:
>
> |             | Huang et al., (2023) | Yuan et al., (2018) | Our Dataset |
> |-------------|---------------------|-----------------------|-------------|
> | Data Source | Based on a work by Moosavi et al. (2019), sourced from Microsoft Bing Map Traffic | Department of Transportation’s official accident reports | Department of Transportation’s official accident reports |
> | Coverage    | 2.8 million accident records, across 13 states, for 2016-2021 | only for Iowa, for 2006-2013 | 9 million records, across 8 states, for a maximum of 20 years up to 2023 |
> | Prediction Tasks | Node-level classification, evaluated on AUROC | Spatial grid-level regression, evaluated under MAE | Edge-level classification/regression, evaluated under both MAE/AUROC |
> | Features    | Road network features | Averaged grid-level traffic volume, rainfall features alongside the grid-level features | Traffic volume and weather feature alongside the road network features |
>
> One can see that our dataset provides more accident records than the two in comparison; In addition, we collect the data directly from official reports, and provide coverage for more years than existing datasets.
>
> There are also many spatiotemporal datasets in existing works; To our knowledge, many existing datasets focus on forecasting traffic speed and volume or are motivated by ride-hailing platforms for demand forecasting. By contrast, our dataset focuses on traffic accident prediction. We think this would be a nice complement to existing works in traffic analysis.
>
> We apologize if this does not come through as clearly in our submission. We will revise the paper to make this comparison more clear.
>
> Yuan, Z., Zhou, X., & Yang, T. (2018, July). Hetero-convlstm: A deep learning approach to traffic accident prediction on heterogeneous spatio-temporal data. KDD.
>
> Huang, B., Hooi, B., & Shu, K. (2023). TAP: A Comprehensive Data Repository for Traffic Accident Prediction in Road Networks. arXiv preprint arXiv:2304.08640.
>
> **>> “After citing multiple references, the author does not evaluate or analyze these papers, merely listing them. To improve the understanding of the importance of these references and their relevance to the current research, it would be beneficial to include appropriate evaluations or analyses within the article.”**
>
> Thanks for the kind suggestion. We added a more detailed comparison between our dataset and related datasets now in Appendix C; These are the most related references to our work.
> We also expanded the related work discussions in Appendix D, including more detailed discussions on several related topics, such as spatiotemporal GNN and graph contrastive learning.

---

### Official Review · Reviewer_FmiK · 2023-07-24
**The paper constructed a traffic dataset, including accidental reports, road networks and traffic volume reports, for traffic accident prediction**

**Rating:** 7
**Confidence:** 4
**Clarity:** Yes

**Strengths:**

1.	The task that this dataset is targeted is very important.
2.	The size of the dataset would contribute to the field, especially for researchers who want to train their robust model for the purpose of accident prediction.
3.	The paper conducted experiments on a few well-known GNNs and compared them.

**Additional Feedback:**

All my feedbacks are in the above sessions.

**Correctness:**

The author claims that the proposed dataset is the largest one so far, but I would suggest that they double-check this since another dataset [1] contains data from 49 states, which is much more than the proposed dataset that has only eight states.
[1] Huang et al., TAP: A Comprehensive Data Repository for Traffic Accident Prediction in Road Networks. 2023

**Documentation:**

Yes

**Limitations:**

Yes

**Opportunities For Improvement:**

1.	I think there is still space for the dataset to involve more data as it currently only covers eight states in the US.
2.	If the task is to predict accident occurrences, e.g., over the next months, it’s quite strange to me that GraphSAGE rather than other spatial-temporal models have the best performance.
3.	There is another repository [1] which I think also needs to be discussed in the paper. It seems to have both state, city and even village levels in 49 states, which seems much larger than the proposed dataset.
4.	Minor: The paper should be more careful when using the plural. E.g., “develop” in line 40, but “incorporates” in line 37, “collects” in line 41, “develops” in line 42.

[1] Huang et al., TAP: A Comprehensive Data Repository for Traffic Accident Prediction in Road Networks. 2023

**Relation To Prior Work:**

The paper should add a discussion regarding [1]
[1] Huang et al., TAP: A Comprehensive Data Repository for Traffic Accident Prediction in Road Networks. 2023

**Summary And Contributions:**

The paper claims that it constructed a traffic dataset to predict traffic accidents. The task itself is very important and can contribute to many real-world scenarios. The dataset involves more than 9 million records, including accidents, road networks, and traffic volume reports. Additionally, based on the constructed dataset, the paper evaluates existing deep-learning models and sets up the benchmark.

---

> ### Author Response · Authors · 2023-08-22
> **Response to Reviewer FmiK 1/2**
>
> Thanks for the reviewer’s detailed comments and feedback on our paper. We are glad that the reviewer appreciates several strengths of our work, such as the importance of our work. Below we respond to each of the comments for improvement and the changes we have made to our datasets and our paper.
>
> **>>> “1. I think there is still space for the dataset to involve more data as it currently only covers eight states in the US.”**
>
> We thank the reviewer for bringing this up; Currently, we have the road network features and the weather information for all the states in the US. The bottleneck is the traffic volume and the accidents. For the eight states in our current dataset, both of these two types of data are published by the Department of Transportation on the respective state’s website; See the links to each state’s government website in Table 3, Appendix A.
>
> For the other states, we have checked their Department of Transportation websites, and we could not find detailed data, including accidents and traffic volume (like the eight states we currently have). Once the data is updated, we would be happy to update our dataset as well.
>
> For a few states, for example, California and New York, the traffic volume data and accident information are both available for a few counties through their county’s transportation departments, such as Los Angeles and New York City. For New York City, we have collected 2.02 million accident records from 2012 to 2023, including the latitude and longitude of each accident. For California, we have 0.4 million Motor Vehicle Crashes from 2016 to 2021, although these do not have the latitude and longitude information, so we cannot match a record to a particular edge/node of the network. We will add these records to our dataset after the review process to facilitate future work.
>
> **>>> “2. If the task is to predict accident occurrences, e.g., over the next months, it’s quite strange to me that GraphSAGE rather than other spatial-temporal models have the best performance.”**
>
> Thanks for the comment. We clarify that we are merely saying that GraghSAGE achieves comparable performance to even some of the more complex models on our dataset, we are not saying that it is strictly the best among all.
>
> Having clarified that, we think one explanation for our results is that spatiotemporal GNNs have several times more trainable parameters than GraphSAGE; thus, they need more training labels to fit the model. On the other hand, the labeling rate of our dataset, i.e., the percentage of edges with a positive label/accident occurrence, is around or below 0.2%. See the statistics below (copied from Table 2):
>
> | States         | DE    | IA    | IL    | MA    | MD    | MN    | MT    | NV    |
> |----------------|-------|-------|-------|-------|-------|-------|-------|-------|
> | Labeling Rate  | 0.23  | 0.07  | 0.14  | 0.10  | 0.15  | 0.05  | 0.05  | 0.12  |
>
> To reinforce our hypothesis above, we additionally trained three spatiotemporal GNNs on our datasets, including STGCN, Graph WaveNet, and AGCRN.  We provide the results below, including the number of trainable parameters in each GNN. We note that the comparisons between GraphSAGE and the other models are generally mixed, with none of them strictly dominating each other. However, GraphSAGE has the lowest number of parameters, and its performance is still comparable to (if not outperforming) other models.
>
> | Model (# parameters) | DE           | MA           | MD           | NV           |
> |--------------------------|--------------|--------------|--------------|--------------|
> | Labeling Rate  | 0.23  |  0.10  | 0.15  | 0.12  |
> | Training records         | 112,670      | 1,265,895    | 341,902      | 101,975      |
> | | | | | |
> | GraphSAGE (115K)    | 87.6±0.1     | 81.8±0.1     | 87.5±0.0     | 91.6±0.9     |
> | AGCRN (219K)           | 86.0±0.2     | 82.1±0.2     | 88.5±0.1     | 90.7±0.2     |
> | STGCN (638K)           | 85.4±0.1     | 81.9±0.3     | 88.7±0.1     | 91.5±0.3     |
> | Graph WaveNet (1079K)  | 85.0±0.2     | 81.9±0.5     | 87.9±0.1     | 90.6±0.2     |
> | DCRNN (1388K)         | 81.2±1.2     | 70.5±0.1     | 84.5±0.3     | 90.5±0.7     |
>
> We apologize for the confusion about our result, and we will revise the paper to make this clear.

---

> ### Author Response · Authors · 2023-08-22
> **Response to Reviewer FmiK 2/2**
>
> **>>> “3. There is another repository [1] which I think also needs to be discussed in the paper. It seems to have both state, city and even village levels in 49 states, which seems much larger than the proposed dataset.”**
>
> We are aware of this online manuscript and have carefully checked the paper in detail before submitting our paper. First, both our work and Huang et al. (2023) (and an earlier work by Yuan et al. (2018)) are targeted at large-scale traffic accident prediction. All three works use OpenStreetMaps to construct road network features. The list of road network features has a large overlap with Huang et al. (2023), including things like the length of a road and the type of a road (e.g., highway or residential, one-way), although we add weather information and traffic volume (AADT labels).
>
> The difference is in the construction of the accident information:
> - Based on the writing of Huang et al. (2023), their dataset uses the accident information collected from another work by Moosavi et al. (2019), which is sourced from Microsoft Bing Map Traffic. Their dataset includes a total of 2.8 million accident records over 5 years (2016-2021), based on Page 4 of their paper.
> - Our data is collected from the Department of Transportation’s official accident reports, which we consider to be more reliable than the above data source. Our dataset includes a total of 9 million records for a maximum of 20 years, e.g., for Massassuchetts, we have records from 2002 to 2023).
> - We map each accident record (latitude and longitude) to the nearest edge, leading to an edge-level classification/regression problem, whereas TAP maps each record to the nearest node, resulting in a node-level classification setting. Thus, our work can be evaluated under both MAE/AUROC metrics, whereas their work is evaluated on the AUROC metric.
>
> Here is a summary table, now added to Appendix D of the updated supplementary materials. We apologize for not being explicit about this comparison in the first place, and we will make it more clear in the revised paper.
>
> |     | **Huang et al. (2023)**       |      **Our Work**   |
> | ------------------------- | ------------------------------------------------------------ | ------------------------------------------------------------ |
> | **Accident Data Source**  | Based on another work by Moosavi et al. (2019), sourced from Microsoft Bing Map Traffic | Department of Transportation’s official accident reports |
> | **Coverage**              | 2.8 million accident records for 2016-2021 | 9 million records for a maximum of 20 years up to 2023 |
> | **Prediction Tasks**      | Node-level classification, evaluated on AUROC | Edge-level classification/regression, evaluated under both MAE/AUROC |
> | **Road Network Features** | Road network features        | Traffic volume and weather feature alongside road network features |
>
> **>>> “4. Minor comments”**
>
> Thanks for noticing these issues! We have revised them accordingly and thoroughly checked the entire paper to fix similar occurrences like the one you pointed out.

---

### Official Review · Reviewer_ZTKf · 2023-07-25
**Reviews**

**Rating:** 6
**Confidence:** 4
**Correctness:** Yes
**Clarity:** Yes

**Strengths:**

The authors construct a large-scale, unified dataset of traffic accident records from various states in the US, totaling 97 million records, which is the largest dataset of this kind to their knowledge. This dataset provides a valuable resource for future research in this area.

The authors find that graph neural networks can accurately predict the number of accidents on roads and whether an accident will occur or not. This suggests that graph neural networks are a promising approach for road safety modeling.

The authors develop multitask and transfer learning techniques to improve the accuracy of the predictions. These techniques leverage the relationships between different tasks and improve the generalization of the model.

**Additional Feedback:**

The paper could be enhanced by incorporating two potential suggestions. Firstly, the addition of a detailed discussion of state-of-the-art spatio-temporal graph learning frameworks enhanced by self-supervised learning would provide a better understanding of the current state of the field and underline the benefits of incorporating self-supervised learning into the proposed approach. The inclusion of relevant research works would also help to contextualize the proposed approach and highlight its unique contributions. Secondly, the paper could be improved by adding a detailed parameter tuning section to evaluate existing solutions on the new dataset. This would improve the reproducibility, comparability, and generalizability of the results, providing valuable insights for future research and facilitating the adoption of the proposed approach. By addressing these limitations, the paper could enhance its quality and contribute to advancing the field of spatio-temporal graph learning.

**Documentation:**

Yes

**Limitations:**

The paper has two potential limitations that could be addressed to improve its quality. Firstly, the authors could include a detailed discussion of state-of-the-art spatio-temporal graph learning frameworks enhanced by self-supervised learning, which would provide a comprehensive analysis of the current state of the field and highlight the potential benefits of incorporating self-supervised learning into their approach. Secondly, the paper could be improved by adding a detailed parameter tuning section for evaluating existing solutions on the new dataset. By addressing these limitations, the paper could provide a better understanding of the proposed approach's performance and potential applications, enhancing its quality and value for future research.

**Opportunities For Improvement:**

One suggestion for improving this work would be to include a detailed discussion of state-of-the-art spatio-temporal graph learning frameworks enhanced by self-supervised learning. Self-supervised learning has emerged as a powerful technique for learning representations from unlabeled data, and it has been successfully applied to various domains.

In the context of spatio-temporal graph learning, self-supervised learning can be used to learn meaningful representations of the graph structure and the temporal dynamics of the data. These approaches have been shown to improve the performance of spatio-temporal graph learning tasks, such as traffic prediction, human activity recognition and crime prediction. By including a detailed discussion of state-of-the-art spatio-temporal graph learning frameworks enhanced by self-supervised learning, the authors could provide a more comprehensive analysis of the current state of the field and highlight the potential benefits of incorporating self-supervised learning into their approach. Some recently proposed relevant research works may include:
"Forecasting fine-grained urban flows via spatio-temporal contrastive self-supervision", TKDE
"Automated spatio-temporal graph contrastive learning", WWW

One suggestion for improving this paper would be to add a detailed parameter tuning section for evaluating existing solutions on this new dataset. The authors evaluate the performance of various deep learning methods on their dataset, but they do not provide a detailed analysis of the hyperparameters used in these methods. Overall, adding a detailed parameter tuning section would enhance the reproducibility and comparability of the results and provide valuable insights for future research in this area.



**Relation To Prior Work:**

Yes

**Summary And Contributions:**

This work presents a study on the use of graph neural networks for predicting traffic accidents on road networks. The authors construct a large-scale, unified dataset of traffic accident records from various states in the US and evaluate the performance of various deep learning methods on this dataset. They find that graph neural networks can accurately predict the number of accidents on roads and whether an accident will occur or not. The authors also develop multitask and transfer learning techniques to improve the accuracy of the predictions. Overall, this work provides insights into the use of graph neural networks for road safety modeling and presents a valuable dataset and code for future research in this area.

---

> ### Author Response · Authors · 2023-08-22
> **Response to Reviewer ZTKf**
>
> Thanks for your kind review of our paper. We are very glad that the reviewer appreciated several strengths of our work, including the timely nature of this study and the extensive dataset we are collecting. We have taken the reviewer’s suggestions for improvements into the revision, and here is our response.
>
> **>>> "Adding a detailed hyper-parameter tuning section."**
>
> Thanks for this suggestion; We have added a section to give more detailed guidance about the hyper-parameters used in our experiments, including the number of layers, the hidden dimensionality of a GNN layer, the learning rate, and the number of training epochs.
>
> To give some examples, we vary the above hyper-parameters and report the ablation studies for two GNNs in terms of validation AUROC scores. We vary one hyper-parameter at a time and keep the others unchanged.
> |                      |  GraphSAGE   |              |              |    DCRNN     |              |              |
> | :-------------------: | :----------: | :----------: | :----------: | :----------: | :----------: | :----------: |
> |   Number of layers    |      2       |      3       |      4       |      2       |      3       |      4       |
> |                       | **85.2**±0.1 |   84.9±0.3   |   84.4±0.4   | **67.8**±1.2 |   67.2±0.8   |   67.3±0.5   |
> | Hidden dimensionality |     128      |     256      |     512      |     128      |     256      |     512      |
> |                       |   84.5±0.4   | **85.2**±0.1 |   84.5±0.5   |   66.9±0.7   | **67.8**±1.2 |   66.9±1.1   |
> |     Learning rate     |   $1e^{-2}$    |   $1e^{-3}$    |   $1e^{-4}$    |   $1e^{-2}$    |   $1e^{-3}$    |   $1e^{-4}$    |
> |                       |   85.0±0.7   | **85.2**±0.1 |   84.0±0.5   |   66.8±1.0   | **67.8**±1.2 |   66.5±0.9   |
> |        Epochs         |      50      |     100      |     200      |      50      |     100      |     200      |
> |                       |   84.0±0.2   | **85.2**±0.1 | **85.2**±0.3 |   66.4±0.7   | **67.8**±1.2 | **67.8**±1.0 |
>
> From the above table, we find that setting the number of layers as 2, hidden dimensionality as 256, and learning rate as 0.001 yields the best results for both cases. The validation performance peaks after training the model for 100 epochs. We also find that these hyper-parameters are useful for other baseline models. Thus, we set them as the default parameters in our package. These results have been added in Appendix E.1 of our newly uploaded supplementary materials.
>
> **>> “Detailed discussion of SoTA spatio-temporal graph learning frameworks enhanced by self-supervised learning.”**
>
> Thanks for your suggestion; It is not hard to add self-supervised learning to our package, and we give an example of implementing graph contrastive learning (You et al. (NeurIPS 2020)) below, by modifying the training loss in the predefined training logic of our package:
>
> ``` Python
> >>> from ml_for_road_safety import Trainer
> # Define a trainer for contrastive learning inherited from the base Trainer class
> >>> class GraphContrastiveLearningTrainer(Trainer):
> # Modify the training loss in the training logic
> >>>     def train_epoch(self):
> #           Define the contrastive loss
> >>>         …
> >>>         loss = info_nce(outputs_1, outputs_2)
> >>>         ...
> # Initialize a contrastive learning trainer
> >>> trainer = GraphContrastiveLearningTrainer(model, dataset, evaluator, ...)
> # Conduct training and evaluation inside the trainer
> >>> log = trainer.train()
> ```
>
> Here are our preliminary results from applying graph contrastive learning to GraphSAGE, DCRNN, and STGCN. We find that graph contrastive learning can indeed improve test performance sometimes, but more evaluations need to be done to draw a conclusion, and this would be a nice direction for future research.
>
> |                  | DE           | MA           | MD           | NV           |
> | ---------------- | ------------ | ------------ | ------------ | ------------ |
> | GraphSAGE        | 87.6±0.1     | 81.8±0.1     | 87.5±0.0     | 91.6±0.9     |
> | GraphSAGE w/ GCL | 86.7±0.2     | **82.4**±0.8 | 85.9±0.4     | **91.8**±0.4 |
> | DCRNN            | 81.2±1.2     | 70.5±0.1     | 84.5±0.3     | 90.5±0.7     |
> | DCRNN w/ GCL     | **86.6**±0.3 | **82.5**±0.7 | **87.8**±0.9 | **91.7**±0.4 |
> | STGCN            | 85.4±0.1     | 81.9±0.3     | 88.7±0.1     | 91.5±0.3     |
> | STGCN w/ GCL     | **86.0**±0.1 | 81.7±0.1 | **89.7**±0.5 | **92.4**±0.1 |
>
> We have also extended the related work to discuss the two references you mentioned. These are added in Appendix D.2 and E.3 of the updated supplementary materials.
>
> You, Yuning, et al. "Graph contrastive learning with augmentations." Advances in neural information processing systems 33 (2020): 5812-5823.

---

### Decision · Program_Chairs · 2023-09-22

**Decision:**

Accept (Poster)

**Comment:**

This paper presents a useful study on graph neural networks' efficacy in predicting traffic accidents, offering a significant dataset of US accident records and valuable insights using multitask and transfer learning techniques. However, the authors could improve this work by incorporating an in-depth review of state-of-the-art spatio-temporal graph learning frameworks enhanced by self-supervised learning, and providing detailed parameter tuning evaluations for existing solutions on their new dataset. Addressing these areas would provide a more comprehensive understanding of the field and enhance the results' reproducibility and comparability.